# DiffusionBlocks: Block-wise Neural Network Training via Diffusion Interpretation

**Makoto Shing**[1]**, Masanori Koyama**[2]**, Takuya Akiba**[1]

[1]Sakana AI, [2]The University of Tokyo

{mkshing,takiba}@sakana.ai, masanori.koyama@weblab.t.u-tokyo.ac.jp

## Abstract

End-to-end backpropagation requires storing activations throughout all layers, creating memory bottlenecks that limit model scalability. Existing block-wise training methods offer means to alleviate this problem, but they rely on ad-hoc local objectives and remain largely unexplored beyond classification tasks. We propose *DiffusionBlocks*, a principled framework for transforming transformer-based networks into genuinely independent trainable blocks that maintain competitive performance with end-to-end training. Our key insight leverages the fact that residual connections naturally correspond to updates in a dynamical system. With minimal modifications to this system, we can convert the updates to those of a denoising process, where each block can be learned independently by leveraging the score matching objective. This independence enables training with gradients for only one block at a time, thereby reducing memory requirements in proportion to the number of blocks. Our experiments on a range of transformer architectures (vision, diffusion, autoregressive, recurrent-depth, and masked diffusion) demonstrate that DiffusionBlocks training matches the performance of end-to-end training while enabling scalable block-wise training on practical tasks beyond small-scale classification. DiffusionBlocks provides a theoretically grounded approach that successfully scales to modern generative tasks across diverse architectures.

Code is available at: https://github.com/SakanaAI/DiffusionBlocks.

## 1 Introduction

**The memory bottleneck in neural network training.** Modern AI led by generative models (Brown et al., 2020; Rombach et al., 2022; Touvron et al., 2023; Peebles & Xie, 2023) has become integral to everyday life. These models rely on *end-to-end backpropagation*, which requires storing intermediate activations across network layers during training. This fundamental requirement causes memory consumption to grow linearly with network depth, creating computational bottlenecks that limit both research flexibility and practical deployment.

**Block-wise training: promises and limitations.** *Block-wise training* methods[1] partition networks into smaller components that can be trained independently, promising dramatic memory savings. Despite this potential, existing approaches (Hinton, 2022; Bengio et al., 2006; Nøkland & Eidnes, 2019; Belilovsky et al., 2019; Siddiqui et al., 2024) consistently underperform end-to-end training. The core challenge is twofold: (1) lack of theoretical grounding: existing methods rely on ad-hoc local objectives without principled coordination between blocks, (2) limited applicability, where they require paradigm-specific designs, task-specific objectives that do not naturally extend beyond classification. Their results are typically demonstrated only on custom architectures without providing systematic procedures to be applied to modern architectures such as Transformers (Vaswani et al., 2017) (Section 4), leaving their applicability to modern generative AI largely unexplored. Without a systematic framework grounded in theory, block-wise training remains an unfulfilled promise.

---

[1]We use *block-wise training* to encompass all approaches that partition networks into independently trainable components. This includes *layer-wise training* as the special case where each block contains one layer.

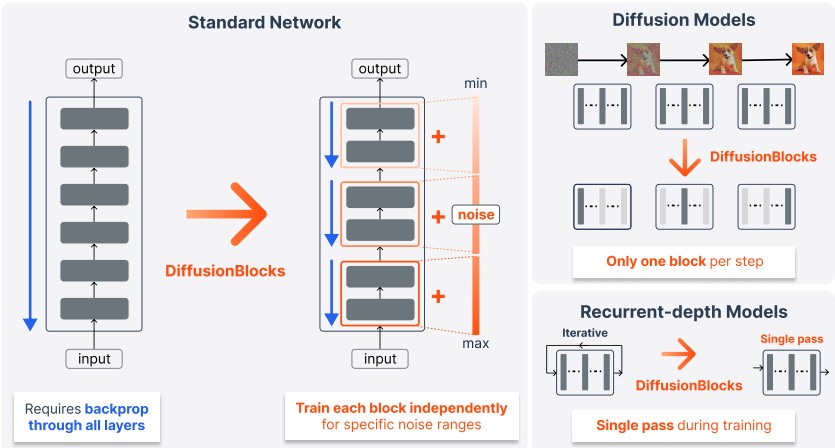

Figure 1: **Overview of DiffusionBlocks. Left:** Standard networks require backpropagation through all layers. **Center:** DiffusionBlocks partitions networks into blocks, each trained independently to denoise within assigned noise ranges. **Right:** Applications. For diffusion models (top), inference requires only the relevant block per denoising step. For recurrent-depth models (bottom), our framework replaces iterative training with single-pass training, eliminating the computational overhead of backpropagation through time.

**Diffusion models: a mathematical foundation for decomposition.** Score-based diffusion models (Song & Ermon, 2019; Song et al., 2021b) model the data distribution through a continuous-time process that gradually adds noise, then learns to reverse this process by estimating the score function at each noise level. Crucially, the denoising step at each noise level can be optimized independently from other noise levels. This independence property provides the theoretical foundation that has been missing from block-wise training approaches: it allows us to partition networks into blocks, each responsible for a specific noise level range, without compromising global coherence.

**Our approach: interpreting networks as diffusion processes.** We propose **DiffusionBlocks**, a framework that enables principled block-wise training by interpreting sequential layer updates in transformer-based networks as discretized steps of a continuous-time diffusion process. Building on the established connection between residual networks and differential equations (Haber & Ruthotto, 2017; Chen et al., 2018), we leverage the fact that residual connections naturally correspond to Euler discretization of the probability flow ODE in diffusion models. This correspondence allows us to partition networks with residual connections, particularly transformer-based networks, into blocks that each handle specific noise-level ranges. These blocks can be trained completely independently, requiring gradients for only one block at a time. Figure 1 illustrates the core concept of Diffusion-Blocks. Unlike previous block-wise methods with ad-hoc objectives, our framework derives each block's training objective from score matching theory. As a result, consistent local optimization at each noise level collectively yields a faithful approximation of the global reverse process, while also allowing practitioners to seamlessly adopt techniques such as those of Karras et al. (2022) to further enhance training.

Our main contributions are:

- **Block-wise training via continuous-time diffusion interpretation:** We show that transformer-based networks can be interpreted as implementing discretized steps of continuous-time diffusion processes (Section 2.2), enabling genuinely independent block training. Each block learns to denoise within its assigned noise level range, requiring gradients for only one block at a time during training (Section 3.1).

- **Equi-probability partitioning for balanced learning**: We propose a principled, diffusion theoretic strategy that partitions noise levels based on equal cumulative probability mass, ensuring balanced parameter utilization across blocks (Section 3.3).

- **Broad applicability with maintained performance:** We conduct extensive experiments (Section 5), demonstrating that DiffusionBlocks successfully applies to diverse architectures (vision, diffusion, autoregressive, recurrent-depth, and masked diffusion), achieving competitive performance to end-to-end backpropagation while requiring gradients for only one block at a time. Additionally, our framework naturally extends to recurrent-depth models, transforming their multiple-iteration training into single-pass training (Section 5.5).

- **Significant efficiency gains:** During training, only one block requires gradient computation, reducing memory requirements proportionally to the number of blocks. For diffusion models, inference requires only one relevant block per denoising step (Section 5.2). For recurrent-depth models, our framework eliminates $K$ iterations during training, demonstrating up to $K$-fold reduction in training computation (Section 5.5).

## 2 PRELIMINARIES

### 2.1 SCORE-BASED DIFFUSION MODELS

We adopt the Variance Exploding (VE) formulation (Song et al., 2021b; Karras et al., 2022) where a clean data $\mathbf{y} \sim p_{\text{data}}$ is perturbed with Gaussian noise at noise level $\sigma$: $\mathbf{z}_\sigma = \mathbf{y} + \sigma\boldsymbol{\epsilon}$ where $\boldsymbol{\epsilon} \sim \mathcal{N}(\mathbf{0}, \mathbf{I})$. For generations, we use the deterministic probability flow ODE that reverses the noising process:

$$\frac{\mathrm{d}\mathbf{z}_\sigma}{\mathrm{d}\sigma} = -\sigma\nabla_\mathbf{z}\log p_\sigma(\mathbf{z}_\sigma), \tag{1}$$

where $\nabla_\mathbf{z}\log p_\sigma(\mathbf{z}_\sigma)$ is the score function. Using Tweedie's formula, the score is approximated via a denoiser $D_{\boldsymbol{\theta}}(\mathbf{z}_\sigma, \sigma)$ that predicts clean data from noisy input: $\nabla_\mathbf{z}\log p_\sigma(\mathbf{z}_\sigma) \approx \frac{D_{\boldsymbol{\theta}}(\mathbf{z}_\sigma, \sigma) - \mathbf{z}_\sigma}{\sigma^2}$ (Robbins, 1992; Hyvärinen, 2005; Vincent, 2011). The denoiser is trained by minimizing:

$$\mathcal{L}(\boldsymbol{\theta}) := \mathbb{E}_{\mathbf{z}_0 \sim p_{\text{data}}, \sigma \sim p_{\text{noise}}, \boldsymbol{\epsilon} \sim \mathcal{N}(\mathbf{0}, \mathbf{I})} \left[ w(\sigma) \| D_{\boldsymbol{\theta}}(\mathbf{y} + \sigma\boldsymbol{\epsilon}, \sigma) - \mathbf{y}\|_2^2 \right], \tag{2}$$

where $w(\sigma)$ weights different noise levels and $p_{\text{noise}}$ is the noise level distribution used during training. The choice of $p_{\text{noise}}$ determines which noise levels are emphasized during training. Karras et al. (2022) uses a log-normal distribution to concentrate training on perceptually important intermediate noise levels where image structure emerges. The weighting $w(\sigma)$ is designed to counteract the sampling bias from $p_{\text{noise}}$, ensuring balanced gradient magnitudes across all noise levels (Karras et al., 2022).

### 2.2 RESIDUAL CONNECTIONS AS EULER STEPS OF THE REVERSE DIFFUSION PROCESS

The connection between residual networks and differential equations has been established in prior works (Haber & Ruthotto, 2017; Chen et al., 2018). We extend this perspective to show that residual networks naturally implement discretized steps of the reverse diffusion process. Applying Euler discretization to Eq. (1) with noise levels $\sigma_0 > \sigma_1 > \cdots > \sigma_T$, we define $\Delta\sigma_\ell := \sigma_{\ell-1} - \sigma_\ell > 0$ and obtain:

$$\mathbf{z}_{\sigma_l} = \mathbf{z}_{\sigma_{l-1}} - \Delta\sigma_\ell \cdot \sigma_{\ell-1}\nabla_\mathbf{z}\log p_{\sigma_{\ell-1}}(\mathbf{z}_{\sigma_{\ell-1}}) \tag{3}$$

$$= \mathbf{z}_{\sigma_{\ell-1}} + \frac{\Delta\sigma_\ell}{\sigma_{\ell-1}}\left(\mathbf{z}_{\sigma_{\ell-1}} - D_{\boldsymbol{\theta}}(\mathbf{z}_{\sigma_{\ell-1}}, \sigma_{\ell-1})\right). \tag{4}$$

As has historically been utilized in the development of the networks with sequential updates, this update rule has an affinity with skip connections. In fact, modern architectures such as Transformers (Vaswani et al., 2017) employ residual connections where each block updates its input through an additive transformation: $\mathbf{z}_\ell = \mathbf{z}_{\ell-1} + f_{\theta_\ell}(\mathbf{z}_{\ell-1})$ where $\mathbf{z}_\ell \in \mathbb{R}^d$ denotes the intermediate output of the block $\ell$, and $f_{\theta_\ell}$ is the block transformation parameterized by $\theta_\ell$. This structure appears in ResNets (He et al., 2016), Transformers, and other modern architectures (Peebles & Xie, 2023; Touvron et al., 2023; DeepSeek-AI et al., 2025). This scheme is also used in the recent development of recurrent-depth models (Dehghani et al., 2019; Fan et al., 2025; Geiping et al., 2025), which apply the same network parameters $\boldsymbol{\theta}$ recursively $K$ times: $\mathbf{z}_k = \mathbf{z}_{k-1} + f_{\boldsymbol{\theta}}(\mathbf{z}_{k-1})$ for $k \in [K]$. However, these methods suffer from the expensive *backpropagation through time (BPTT)*, and various measures have been taken to reduce its computational burden, for example, by gradient truncation (Williams & Zipser, 1995; Mikolov et al., 2010; Geiping et al., 2025). That being said, the

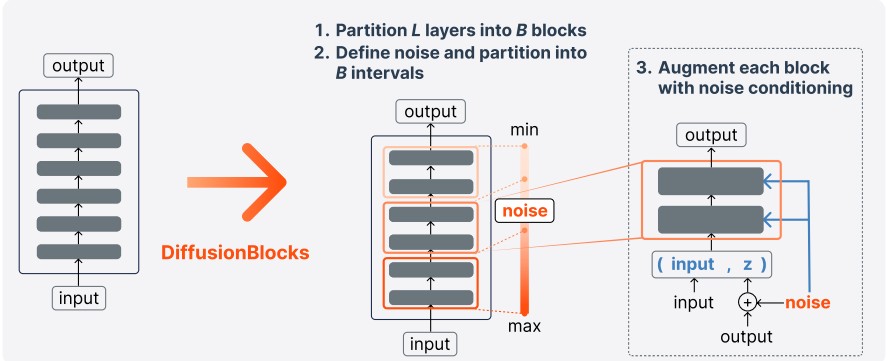

Figure 2: **3-step conversion of a standard neural network to DiffusionBlocks at training phase.**
**Step 1:** Partition $L$ layers into $B$ blocks. **Step 2:** Define noise distribution $p_\sigma$ (e.g., log-normal) and partition the range $[\sigma_{\min}, \sigma_{\max}]$ into $B$ intervals $\{[\sigma_b, \sigma_{b-1}]\}_{b=1}^B$, assigning each block a specific noise range (Section 3.3). **Step 3:** Augment blocks with noise conditioning: extend input to $\tilde{\mathbf{x}} = (\mathbf{x}, \mathbf{z}_\sigma)$ where $\mathbf{z}_\sigma = \mathbf{y} + \sigma\boldsymbol{\epsilon}$, and incorporate noise-level conditioning (e.g., via AdaLN). Then, each block is trained independently from other blocks to predict target $\mathbf{y}$ within its assigned noise range.

---

| Standard Network – Training |
| --- |
| 1: **Given:** Network with parameters $\boldsymbol{\theta}$ |
| 2: Sample data $(\mathbf{x}, \mathbf{y})$ |
| 3: $\mathbf{z}_0 \leftarrow \mathbf{x}$ |
| 4: **for** $\ell = 1$ to $L$ **do** |
| 5: $\quad \mathbf{z}_\ell \leftarrow \mathbf{z}_{\ell-1} + f_{\theta_\ell}(\mathbf{z}_{\ell-1})$ |
| 6: **end for** |
| 7: $\hat{\mathbf{y}} \leftarrow \mathbf{z}_L$ |
| 8: $\mathcal{L} \leftarrow \text{Loss}(\hat{\mathbf{y}}, \mathbf{y})$ |
| 9: Update all $\boldsymbol{\theta}$ via backprop |

| DiffusionBlocks – Training |
| --- |
| 1: **Given:** A single block $b \in [B]$ with parameters $\boldsymbol{\theta}_b$ |
| 2: Sample data $(\mathbf{x}, \mathbf{y})$ |
| 3: Sample $\sigma \sim p_{\text{noise}}^{(b)}$ from $[\sigma_b, \sigma_{b-1}]$ |
| 4: $\hat{\mathbf{y}} \leftarrow \bar{f}_{\boldsymbol{\theta}_b\mid\sigma}(\mathbf{x}, \mathbf{y} + \sigma\boldsymbol{\epsilon})$, where $\boldsymbol{\epsilon} \sim \mathcal{N}(\mathbf{0}, \mathbf{I})$ ▷ Apply block $b$ to denoise |
| 5: $\mathcal{L} \leftarrow w(\sigma) \cdot \text{Loss}(\hat{\mathbf{y}}, \mathbf{y})$ ▷ Weighted loss |
| 6: Update only $\boldsymbol{\theta}_b$ via backprop |

| Standard Network – Inference |
| --- |
| 1: **Input:** $\mathbf{x}$ |
| 2: $\mathbf{z}_0 \leftarrow \mathbf{x}$ |
| 3: **for** $\ell = 1$ to $L$ **do** |
| 4: $\quad \mathbf{z}_\ell \leftarrow \mathbf{z}_{\ell-1} + f_{\theta_\ell}(\mathbf{z}_{\ell-1})$ |
| 5: **end for** |
| 6: **Output:** $\mathbf{z}_L$ |

| DiffusionBlocks – Inference |
| --- |
| 1: **Input:** $\mathbf{x}$, noise levels $\{\sigma_i\}_{i=1}^T$ ▷ Typically, $T = B$ |
| 2: $\mathbf{z}_0 \sim \mathcal{N}(\mathbf{0}, \sigma_{\max}^2 \mathbf{I})$ |
| 3: **for** $i = 0$ to $T - 1$ **do** |
| 4: $\quad$ Select block $b$ where $\sigma_i \in [\sigma_b, \sigma_{b-1}]$ |
| 5: $\quad \hat{\mathbf{y}} \leftarrow \bar{f}_{\boldsymbol{\theta}_b\mid\sigma_{i-1}}(\mathbf{x}, \mathbf{z}_{i-1})$ |
| 6: $\quad \mathbf{z}_i \leftarrow \text{Euler step}(\mathbf{z}_{i-1}, \hat{\mathbf{y}}, \sigma_{i-1}, \sigma_i)$ ▷ Eq. (5) |
| 7: **end for** |
| 8: **Output:** $\mathbf{z}_T$ |

Figure 3: **Training and inference algorithms for standard residual networks (left) versus DiffusionBlocks (right).** Given: A $L$-layer network partitioned into $B$ blocks with noise ranges $\{[\sigma_b, \sigma_{b-1}]\}_{b=1}^B$, noise distribution $p_\sigma$, and training data $\{(\mathbf{x}_n, \mathbf{y}_n)\}_{n=1}^N$. The function $w(\sigma)$ denotes the loss weighting, and $\bar{f}_{\boldsymbol{\theta}_b\mid}$ represents the noised-conditioned block with parameters $\boldsymbol{\theta}_b$.

critical observation is that, in the setting of the diffusion introduced in the previous section, $D_{\boldsymbol{\theta}}$ itself in Eq. (4) can be trained with Eq. (2) without BPTT, thereby providing a theoretically sound optimization method of a dynamical system through an ensemble of local optimization. In the next section, we provide a recipe for converting networks with skip connections into diffusion, thereby replacing the *backpropagation through layers* with the optimization scheme analogous to Eq. (2).

# 3 METHOD

## 3.1 CONVERTING A NEURAL NETWORK TO DIFFUSIONBLOCKS

Our goal in this section is to transform a given feedforward system into a discretized version of the recursive denoising steps in the diffusion model. Throughout this paper, we denote by $(\mathbf{x}, \mathbf{y})$ the input-output pairs where $\mathbf{x}$ represents the network input (e.g., images for classification) and $\mathbf{y}$ is the target output (e.g., class label for classification). Figure 1 provides an overview: instead of backpropagating through all layers, we partition networks into blocks that independently learn to denoise within assigned noise level ranges. Consider a neural network in a form of a stack of set-to-set maps (e.g. transformer-based networks) $\mathcal{F} = \{f_{\theta_\ell} \mid \ell \in [L]\}$ with the same output and input dimensions, so that $f_{\theta_\ell}$ maps a variable set of tokens in $\mathbb{R}^d$ to the same number of tokens in $\mathbb{R}^d$. The original network therefore processes the input with $f_{\theta_L} \circ \cdots \circ f_{\theta_0}$, followed possibly by a readout module. Or, in more conventional formulation with the presence of residual, the original network may update the $\ell$-th layer input $\mathbf{z}_\ell$ to the next layer via the rule $\mathbf{z}_{\ell+1} = \mathbf{z}_\ell + f_{\theta_\ell}(\mathbf{z}_\ell)$. We transform this network into a stack of Diffusion Blocks through the following three steps (Figure 2).

**Step 1: Block partitioning.** We partition $\mathcal{F}$ into $B$ blocks $\mathcal{F} = \uplus_{b=1}^{B} \mathcal{F}_b$, where $\mathcal{F}_b$ contains layers indexed by $\{\ell_{b-1} + 1, \ldots, \ell_b\}$. Let $\bar{f}_{\boldsymbol{\theta}_b} := f_{\theta_{\ell_b}} \circ \cdots \circ f_{\theta_{\ell_{b-1}+1}}$ be the composition of layers in $\mathcal{F}_b$.

**Step 2: Noise range assignment.** We define a noise distribution $p_{\text{noise}}$ and define a noise range $[\sigma_{\min}, \sigma_{\max}]$. We partition the range into $B$ intervals $\{[\sigma_b, \sigma_{b-1}]\}_{b=1}^{B}$. We recommend the choice of log-normal for $p_{\text{noise}}$, following Karras et al. (2022), along with the partitioning strategy in Section 3.3.

**Step 3: Augmenting blocks with noise conditioning.** Finally, we suit $\{\bar{f}_{\boldsymbol{\theta}_b}\}_b$ to the update rule in Eq. (4) by letting $\bar{f}_{\boldsymbol{\theta}_b}$ play the role of $D_{\boldsymbol{\theta}_b}$. Leveraging the assumption that $\bar{f}_{\boldsymbol{\theta}_b}$ is a map from a set of tokens to a set of tokens, we alter the input $\bar{f}_{\boldsymbol{\theta}_b}$ from $\mathbf{x}$ to $\tilde{\mathbf{x}} = (\mathbf{x}, \mathbf{z})$. Additionally, we extend each block $f_{\boldsymbol{\theta}_b}$ to incorporate noise-level conditioning through, for example, via normalization (AdaLN) (Peebles & Xie, 2023). We denote this noise-conditioned version as $\bar{f}_{\boldsymbol{\theta}_b|\sigma}$. Altogether, the update of the diffusion block constructed from $\mathcal{F}$ is given by:

$$\mathbf{z}_b = \mathbf{z}_{b-1} + \frac{\Delta \sigma_b}{\sigma_{b-1}} \left( \mathbf{z}_{b-1} - [\bar{f}_{\boldsymbol{\theta}_b|\sigma_{b-1}}(\mathbf{x}, \mathbf{z}_{b-1})]_{\mathbf{z}} \right), \tag{5}$$

where $[\bar{f}(\cdot)]_{\mathbf{z}}$ is the set of tokens corresponding to $\mathbf{z}$ (i.e. $\bar{f}(\cdot) = ([\bar{f}(\cdot)]_{\mathbf{x}}, [\bar{f}(\cdot)]_{\mathbf{z}})$). More abstractly put, our modified update rule Eq. (5) can be rewritten as $\mathbf{z}_b = \alpha \mathbf{z}_{b-1} + \beta \bar{f}_{\boldsymbol{\theta}_b|\sigma_{b-1}}(\mathbf{x}, \mathbf{z}_{b-1})$ where $\alpha$ and $\beta$ are constants dependent on $\sigma$ ratio. We note that our modification of the network into the stack of diffusion blocks maintains most of the structure of the original, particularly in the presence of skip connection, so that $\mathbf{z}_\ell = \mathbf{z}_{\ell-1} + f_{\theta_\ell}(\mathbf{z}_{\ell-1})$ is the original update rule. At the time of inference, $\mathbf{z}_b$ serves as the intermediate estimator of the target variable, with $\mathbf{z}_0 = \sigma_{\max}\epsilon$ being the pure noise. Please see Figure 6 in Appendix B for the conversion of this inference process.

## 3.2 BLOCK-INDEPENDENT TRAINING OF THE DIFFUSION BLOCKS

By the network modification recipe in the previous section, we transform the original feedforward map to the recursive denoising map in a diffusion process. The advantage of this modification is the fact that the objective in Eq. (2) can be optimized at any noise level $\sigma$ independently without knowledge of other noise levels. This allows us to define a training objective for each block $b$:

$$\mathcal{L}_b(\boldsymbol{\theta}_b) := \mathbb{E}_{(\mathbf{x},\mathbf{y}) \sim p_{\text{data}}, \sigma \sim p_{\text{noise}}^{(b)}, \epsilon \sim \mathcal{N}(\mathbf{0},\mathbf{I})} \left[ w(\sigma) \cdot \text{Loss}(\bar{f}_{\boldsymbol{\theta}_b|\sigma}(\mathbf{x}, \mathbf{y} + \sigma\epsilon), \mathbf{y}) \right], \tag{6}$$

where $p_{\text{noise}}^{(b)}$ is the noise distribution $p_{\text{noise}}$ with the support of $[\sigma_b, \sigma_{b-1}]$ and renormalized, and $\text{Loss}(\cdot, \cdot)$ is the inner loss function, typically L2 loss as in Eq. (2). Each block independently learns to denoise within its assigned range, with training samples drawn according to the original distribution $p_{\text{noise}}$. Collectively, the $B$ blocks cover the entire noise distribution: $\bigcup_{b=1}^{B} [\sigma_b, \sigma_{b-1}] = [\sigma_{\min}, \sigma_{\max}]$, ensuring that the complete network can denoise at any noise level while each block specializes in its designated range. This independence enables training with memory requirements for only $L/B$ layers, storing activations only for the active block, compared to all $L$ layers required by

standard training. More succinctly in comparison to the original network, we gain this block-wise independence from the fact that $\bar{f}_{\boldsymbol{\theta}_b|\sigma}(\mathbf{x}, \mathbf{y} + \sigma\boldsymbol{\epsilon})$ is now modified to predict $\mathbf{y}$ for each $b$. This way, training for each block can be carried out without waiting to receive the output of the previous layer. Please see Figure 5 in Appendix B for the training process in the specific adaptations for different architectures. Figure 3 provides an algorithmic procedure of training and inference. This approach achieves a $B\times$ memory reduction during training, as gradients are computed for only one block at a time.

### 3.3 BLOCK PARTITIONING STRATEGY

A critical design choice in DiffusionBlocks is how to partition the noise level range $[\sigma_{\min}, \sigma_{\max}]$ into $B$ intervals. A naive approach would divide the range uniformly: $\sigma_b = \sigma_{\min} + b \cdot (\sigma_{\max} - \sigma_{\min})/B$. However, this fails to account for the varying difficulty of denoising at different noise levels. Following Karras et al. (2022), we adopt a log-normal distribution for sampling noise levels during training: $\log \sigma \sim \mathcal{N}(P_{\text{mean}}, P_{\text{std}}^2)$. This distribution concentrates probability mass at intermediate noise levels, which empirically contribute most to generation quality.

To preserve this distribution across the entire network while ensuring each block handles equal denoising difficulty, we partition based on cumulative probability mass. Specifically, we choose boundaries $\{\sigma_b\}_{b=1}^B$ such that each block handles exactly $1/B$ of the total probability mass: $\int_{\sigma_{b-1}}^{\sigma_b} p_{\text{noise}}(\sigma)d\sigma = 1/B$. The block boundaries are computed as $\sigma_b = \exp(P_{\text{mean}} + P_{\text{std}} \cdot \Phi^{-1}(q_b))$, where $\Phi^{-1}$ is the inverse standard normal CDF and $q_b = q_{\min} + \frac{b}{B}(q_{\max} - q_{\min})$, with $q_{\min/\max} = \Phi\left(\frac{\log \sigma_{\min/\max} - P_{\text{mean}}}{P_{\text{std}}}\right)$. This *equi-probability partitioning* ensures that each block handles an equal amount of the training distribution's probability mass, leading to balanced parameter utilization. As shown in Figure 4, blocks assigned to intermediate noise levels, where denoising is most challenging, receive narrower intervals, while blocks handling very high or low noise levels receive wider intervals. This strategy optimizes learning efficiency across all blocks. In Section 5.6, we demonstrate that this strategy contributes significantly to the training of DiffusionBlocks. Also, see Appendix C for implementation details.

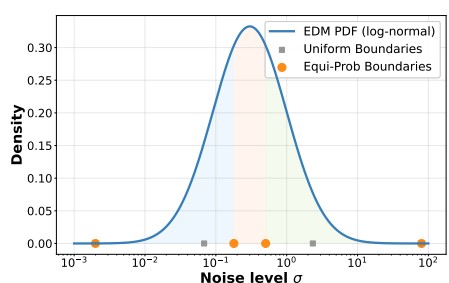

Figure 4: **Equi-probability partitioning** ($B = 3$). Blocks partition the log-normal $p_\sigma$ by equal probability mass (orange boundaries), not uniform spacing (gray), concentrating capacity where denoising is most challenging.

## 4 RELATED WORKS

**Block-wise training methods.** Various block-wise training approaches (Hinton, 2022; Bengio et al., 2006; Nøkland & Eidnes, 2019; Belilovsky et al., 2019; Siddiqui et al., 2024) partition networks into independently trainable components but lack theoretical grounding, relying on heuristic objectives that fail to guarantee global performance when optimized locally. Approaches like Forward-Forward algorithm (Hinton, 2022) rely on contrastive objectives, which fundamentally limit them to classification tasks and make adaptation to generation non-trivial. In contrast, DiffusionBlocks leverages denoising score matching theory, which naturally decomposes into independent local objectives without task-specific constructs, enabling application to both classification and generative tasks.

**Comparison with NoProp.** Concurrently with our submission, Li et al. (2025) has also released a backpropagation-free strategy in close relation to our philosophy. However, they present their technique together with the custom CNN-based architecture in one package and evaluate only on classification tasks, making it unclear how to apply their approach to modern architectures or tasks other than the classification they showcase in their work. In contrast, DiffusionBlocks provides a systematic procedure for converting any residual networks, particularly modern transformers, into block-wise trainable models with minimal modifications. We partition the continuous noise range using equi-probability partitioning and demonstrate success on both generative tasks and classifica-

Table 1: **ViT results on CIFAR-100.** DiffusionBlocks achieves comparable accuracy while training only 4 layers at a time, outperforming Forward-Forward algorithm.

Table 2: **DiT results for image generation.** FID is computed on both training and test splits (train / test). DiffusionBlocks achieves comparable scores while reducing training memory and inference cost by $3\times$.

Table 3: **Masked diffusion model (MDM) results on text8.** DiffusionBlocks improves BPC while training with $3\times$ less memory through masking schedule partitioning.

| Method | Accuracy ($\uparrow$) |
|---|---|
| ViT | 60.25 |
| + Forward-Forward | 7.85 |
| **+ DiffusionBlocks** | **59.30** |

| Dataset | Method | FID ($\downarrow$) |
|---|---|---|
| CIFAR-10 | DiT | 32.84 / 39.83 |
| | **+ DiffusionBlocks** | **30.59 / 37.20** |
| ImageNet | DiT | 9.01 / 12.09 |
| | **+ DiffusionBlocks** | **9.00 / 10.63** |

| Method | BPC ($\downarrow$) |
|---|---|
| MDM | 1.56 |
| **+ DiffusionBlocks** | **1.45** |

tion tasks. In Section 5.6.1, we apply DiffusionBlocks to their architecture, and demonstrate that our continuous-time block-wise training with equi-probability partitioning is more effective.

**Stage-specific diffusion models.** Several works train specialized models for different noise levels in diffusion (Balaji et al., 2023; Fang et al., 2024; Park et al., 2024; Reuss et al., 2025). However, these approaches train models jointly or fine-tune from shared parameters. DiffusionBlocks trains blocks independently, with no shared parameters or joint fine-tuning, achieving complete isolation.

# 5 EXPERIMENTAL RESULTS

We evaluate DiffusionBlocks across diverse architectures and tasks to demonstrate its generality and effectiveness. Detailed experimental configurations are provided in Appendix E. For each architecture, we report task performance alongside the memory reduction factor $B$, where only $L/B$ layers require gradients during training.

**Baselines.** Because DiffusionBlocks is a framework for transforming networks into block-wise trainable models, we evaluate its efficacy by comparing the modified network (trained block-wise) against the original network (trained with end-to-end backpropagation). Other block-wise training methods in practice today also include Forward-Forward (FF) (Hinton, 2022) and the concurrent NoProp (Li et al., 2025). Fair comparison against these methods warrants careful experimental design. Firstly, we compare against FF only on classification tasks (Section 5.1) since its contrastive objective does not naturally extend to generation. Also, because NoProp is proposed together with a custom architectural design rather than with a principled transformation procedure to be applied to a vanilla network, the adaptation of NoProp to other architectures involves nontrivial design choices and freedom. To enable fair comparison with NoProp, we therefore use their specific architecture as the base diffusion model on which to apply our DiffusionBlocks (Section 5.6.1).

## 5.1 VISION TRANSFORMERS FOR IMAGE CLASSIFICATION

We first validate DiffusionBlocks on classification tasks using Vision Transformer (ViT) (Dosovitskiy et al., 2021) on CIFAR-100 (Krizhevsky, 2009). A 12-layer ViT is partitioned into $B$=3 blocks, with noise added to class label embeddings during training. We compare against the Forward-Forward algorithm, a representative block-wise training method that uses contrastive objectives. Table 1 maintains baseline accuracy while requiring gradients for only 4 layers. Notably, Forward-Forward achieves only 7.85% accuracy, highlighting the importance of principled denoising objectives over ad-hoc contrastive approaches.

## 5.2 DIFFUSION MODELS FOR IMAGE GENERATION

Having established its effectiveness on classification tasks, we now turn to generative models. We begin with image generation, where DiffusionBlocks provides both training and inference efficiency benefits. We apply DiffusionBlocks to DiT (Peebles & Xie, 2023) within the EDM (Karras et al., 2022) framework. We evaluate 12-layer DiT (DiT-S/2) on CIFAR-10 (Krizhevsky, 2009) and 24-layer DiT (DiT-L/2) on ImageNet at $256 \times 256$ resolution (Deng et al., 2009), both with $B$=3 blocks. During inference, we use Euler sampling with 50 steps and classifier-free guidance (scale 2.0) (Ho

Table 4: **Autoregressive (AR) transformer results for text generation.** DiffusionBlocks maintains generation quality with $4\times$ memory reduction on both LM1B and Openwebtext (OWT) datasets.

| Dataset | Method | MAUVE ($\uparrow$) | PPL (`Llama-2`) ($\downarrow$) | PPL (`GPT2-XL`) ($\downarrow$) |
|---------|--------|--------------------|-------------------------------|-------------------------------|
| LM1B | AR | 0.50 | 14.58 | 38.87 |
| | + DiffusionBlocks | **0.71** | **12.32** | **30.99** |
| OWT | AR | **0.85** | 15.05 | **25.24** |
| | + DiffusionBlocks | 0.82 | **14.99** | 26.33 |

Table 5: **Recurrent-depth model results for text generation.** DiffusionBlocks eliminates 32 training iterations, achieving better performance with single-pass training.

| Method | MAUVE ($\uparrow$) | PPL (`Llama-2`) ($\downarrow$) | PPL (`GPT2-XL`) ($\downarrow$) |
|--------|--------------------|-------------------------------|-------------------------------|
| `Huginn` (Geiping et al., 2025) | 0.49 | 17.04 | 46.73 |
| **+ DiffusionBlocks** | **0.70** | **16.08** | **42.43** |

& Salimans, 2021). Table 2 shows that DiffusionBlocks achieves comparable FID scores with $3\times$ memory reduction. Additionally, inference requires only one block per denoising step, providing computational savings proportional to the number of steps.

## 5.3 Masked diffusion models for text generation

We extend DiffusionBlocks to masked diffusion language models using MD4 (Shi et al., 2024) on the text8 dataset (Mahoney, 2011). While continuous diffusion models naturally map to our framework through noise levels $\sigma$, extending DiffusionBlocks to discrete masked diffusion requires careful adaptation. Specifically, we partition the masking schedule rather than continuous noise levels, ensuring each block handles an equal share of the demasking work (details in Appendix D). We use a 12-layer DiT-based transformer (Lou et al., 2024; Sahoo et al., 2024) partitioned into $B=3$ blocks. Table 3 shows that DiffusionBlocks achieves 1.45 bits-per-character (BPC) compared to MD4's 1.56, while using $3\times$ less memory. This improvement confirms that our principled noise-level partitioning effectively extends to discrete diffusion processes.

## 5.4 Autoregressive models for text generation

We demonstrate that DiffusionBlocks successfully transforms standard autoregressive (AR) models, which are architectures originally designed for next-token prediction, not denoising. Using 12-layer Llama-2-style transformers (Touvron et al., 2023) with $B=4$ blocks, we evaluate on 1 Billion Words Dataset (LM1B) (Chelba et al., 2014) and OpenWebText (OWT) (Gokaslan & Cohen, 2019). While AR models are typically evaluated using perplexity, computing traditional perplexity is non-trivial for our diffusion framework as it is not derived from ELBO. Instead, we evaluate using MAUVE (Pillutla et al., 2021) scores following SEDD (Lou et al., 2024) to measure similarity between generated and real text. We also report generative perplexity from two teacher models, `Llama-2-7B` and `GPT2-XL` (Radford et al., 2019), following Lou et al. (2024); Sahoo et al. (2024). Table 4 shows that DiffusionBlocks achieves comparable performance despite training only 3 layers at a time, demonstrating the framework's broad applicability beyond diffusion-native architectures.

## 5.5 Recurrent-depth models for text generation

We now showcase a different application of DiffusionBlocks beyond block-wise training. As noted in Section 2.2, the updates in recurrent-depth models naturally correspond to diffusion steps. Following Section 3.1, we apply DiffusionBlocks to `Huginn` (Geiping et al., 2025), a recurrent-depth model that applies the same network multiple times, starting from noise. While `Huginn` uses 8-step truncated BPTT to avoid the full BPTT over 32 iterations, DiffusionBlocks makes this optimization even more efficient, because it only requires a single forward pass per training step. Table 5 shows better performance on LM1B for text generation while eliminating 32 iterations. This demonstrates that our framework enables fundamental training transformations beyond block-wise training.

Table 6: **Comparison with NoProp on CIFAR-100.** DiffusionBlocks achieves both continuous-time formulation and layer-wise training. All scores except DiffusionBlocks are taken from NoProp (Li et al., 2025). Note that the Backprop in this table is the result of applying BPTT to the sampled paths of a specific form of SDE that equationally resembles the process used in `NoProp-DT`. See Li et al. (2025) for the details.

| Method | Continuous | Block-wise | Accuracy ($\uparrow$) |
|---|---|---|---|
| Backprop | | | 47.80 |
| NoProp-DT | | $\checkmark$ | 46.06 |
| NoProp-CT | $\checkmark$ | | 21.31 |
| NoProp-FM | $\checkmark$ | | 37.57 |
| **(Ours) DiffusionBlocks** | $\checkmark$ | $\checkmark$ | **46.88** |

Table 7: **Effect of block partitioning strategy on CIFAR-10.** Layer distribution indicates the number of layers in each of the 3 blocks (totaling 12 layers).

| Partitioning Strategy | Layer Distribution | FID ($\downarrow$) |
|---|---|---|
| Uniform | [4,4,4] | 43.53 |
| Uniform | [3,6,3] | 43.59 |
| Uniform | [6,4,2] | 47.49 |
| Uniform | [2,4,6] | 42.37 |
| Equi-Probability | [4,4,4] | **38.03** |
| Equi-Probability | [3,6,3] | 41.64 |
| Equi-Probability | [6,4,2] | 45.42 |
| Equi-Probability | [2,4,6] | 40.40 |

Table 8: **Effect of block count on ImageNet.** Fewer blocks achieve better FID but require more layers per diffusion step, creating a trade-off between quality and efficiency. The scores that surpass the end-to-end backpropagation ($B$=1) are highlighted in **bold**.

| Number of Blocks | FID ($\downarrow$) | L/B ($\downarrow$) | Relative Speed |
|---|---|---|---|
| $B = 1$ | 12.09 | 24 | 1.0$\times$ |
| $B = 2$ | **9.90** | 12 | 2.0$\times$ |
| $B = 3$ | **11.11** | 8 | 3.0$\times$ |
| $B = 4$ | **11.90** | 6 | 4.0$\times$ |
| $B = 6$ | 14.43 | 4 | 6.0$\times$ |

## 5.6 ANALYSIS

### 5.6.1 COMPARISON WITH NOPROP

We compare DiffusionBlocks with NoProp as an ablation study, applying to their custom CNN-based architecture to isolate the effect of our continuous-time block-wise training using equi-probability partitioning. Table 6 shows results on CIFAR-100 classification (details in Appendix E.6.1). DiffusionBlocks outperforms all NoProp variants. Notably, while maintaining comparable performance to the backpropagation, DiffusionBlocks is the only method that successfully combines continuous-time formulation with block-wise training. This demonstrates that our equi-probability partitioning with independent denoisers per block is crucial for continuous-time block-wise training.

### 5.6.2 ABLATION STUDIES ON DESIGN CHOICES

We conduct ablation studies to analyze key design choices in DiffusionBlocks. All experiments follow the configurations described in Section 5.2.[2]

**Block partitioning strategy.** Table 7 compares our equi-probability partitioning with uniform partitioning on CIFAR-10. Equi-probability partitioning achieves significantly better FID across all layer distributions. The improvement stems from allocating computational resources based on denoising difficulty: equi-probability assigns more blocks to challenging intermediate noise levels where most learning occurs, while uniform partitioning wastes capacity on trivial very high/low noise regions. Notably, within equi-probability partitioning, uniform layer distribution (4-4-4) achieves the best FID, demonstrating that practitioners can simply divide layers equally without tuning since the noise-based partitioning automatically balances learning difficulty across blocks.

**Number of blocks $B$.** Table 8 summarizes the effect of varying the number of blocks on ImageNet (see Appendix F.2 for the results on CIFAR-10). It reveals the trade-off between generation quality and efficiency on ImageNet. Notably, moderate block counts ($B$=2 or $B$=3) achieve better FID than

---

[2]These ablations disable block overlap in Appendix C to isolate the effectiveness of each component, resulting in the FID difference from Table 2.

end-to-end training ($B$=1), suggesting that moderate block partitioning can actually improve performance through specialization. As $B$ increases further, quality gradually declines due to reduced capacity per block, though inference speed improves linearly. The optimal $B$ varies across tasks (see Appendix F.3 for language modeling results).

## 6 CONCLUSION

We introduced DiffusionBlocks, a theoretically grounded framework that transforms residual networks into independently trainable blocks through continuous-time diffusion interpretation. By recognizing that residual connections naturally implement discretized diffusion steps, we provide a systematic recipe requiring minimal modifications that maintains competitive performance across diverse architectures while achieving $B\times$ memory reduction during training.

**Future works.** Our work opens several important directions for future research. First, while we consistently used Euler discretization to match residual connections, other diffusion samplers (Song et al., 2021a; Lu et al., 2023; Zhao et al., 2023) could be employed within blocks with modified interblock connections. Second, DiffusionBlocks currently requires matching input-output dimensions, which limits its application to architectures like U-Net (Ronneberger et al., 2015). Third, while we demonstrate DiffusionBlocks' effectiveness on models trained from scratch, scaling to even larger models would further demonstrate its practical impact. Particularly, a promising direction is to convert pre-trained large models to DiffusionBlocks through fine-tuning rather than training from scratch. Fourth, determining the optimal granularity of block partitioning presents an interesting theoretical and practical challenge. While our experiments demonstrate that treating entire architectural blocks (e.g., complete ViT blocks) as single denoising units works well, a principled method for selecting the ideal partitioning granularity based on architecture and task characteristics could further enhance the framework's applicability. Finally, understanding why moderate block partitioning sometimes outperforms end-to-end training warrants theoretical investigation. We hypothesize two contributing factors: (1) DiffusionBlocks employs a different optimization structure in which each block is directly linked to the target through a denoising objective in Eq.( 13), creating a learning signal that differs from standard end-to-end training; and (2) assigning different noise ranges to different blocks may induce beneficial specialization effects. Combined with equi-probability partitioning, this introduces a natural form of curriculum learning (Bengio et al., 2009) by allocating balanced difficulty across blocks. Developing a formal theory and analysis for these effects could reveal new principles for scalable and structured neural network optimization beyond memory efficiency.

DiffusionBlocks represents a step toward democratizing large-scale model training by reducing computational requirements without sacrificing performance, making advanced AI capabilities more accessible.

## AUTHOR CONTRIBUTIONS

Makoto Shing conceptualized the DiffusionBlocks framework, developed its diffusion-theoretic formulation connecting residual networks and continuous-time diffusion processes, implemented the method, conducted all experiments, and wrote the manuscript. Masanori Koyama provided theoretical insights into the diffusion-based interpretation and contributed to refining both the manuscript and the conceptual positioning of the work. Takuya Akiba supervised the research and provided technical guidance and feedback throughout the project. All authors contributed to the interpretation of results and manuscript revision.

## ACKNOWLEDGEMENT

The authors would like to thank Stefano Peluchetti for helpful feedback on an earlier version of the draft.

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

## A NOTATIONS

In this section, we provide the notations that we will be using in the ensuing mathematical formulations and statements.

## B EXTENSION TO DIVERSE ARCHITECTURES

While we have described DiffusionBlocks for standard residual networks where inputs and outputs naturally live in the same $d$-dimensional space, the framework extends to specialized architectures. Figures 5 and 6 illustrate how different model types can be converted to DiffusionBlocks for training and inference, respectively.

For Vision Transformers (ViT) (Dosovitskiy et al., 2021) in classification tasks (top left), we adapt DiffusionBlocks by adding noise to the class label embeddings while maintaining the standard ViT architecture. Specifically, we create the input sequence by concatenating the `[CLS]` token, patch

| Notation | Description |
|---|---|
| $x \sim \mathcal{X}$ | Conditioning/Input to the network (task-dependent: see below) |
| $y \in \mathcal{Y}$ | Clean target data (task-dependent: see below) |
| $\sigma \in \mathbb{R}$ | Noise level in continuous diffusion. |
| $z_\sigma \in \mathbb{R}^d$ | Noisy data at noise level $\sigma$: $z_\sigma = y + \sigma\epsilon$, where $\epsilon \sim \mathcal{N}(0,1)$ |
| $z_\ell \in \mathbb{R}^d$ | Intermediate activation at layer/block $\ell$ |
| $D_\theta : \mathbb{R}^d \times \mathbb{R} \to \mathcal{Y}$ | Denoiser network with parameters $\theta$ |
| $f_{\theta_\ell} : \mathbb{R}^d \to \mathbb{R}^d$ | Layer/block transformation with parameters $\theta_\ell$ |
| $B$ | Number of blocks |
| $L$ | Total number of layers |

**Examples of $(x, y)$ on a task:**

| | |
|---|---|
| Image classification | $x$: input image, $y$: class label |
| Image generation | $x$: noisy image (optionally, and class label), $y$: clean image |
| Text Generation (AR) | $x$: previous tokens, $y$: next token |
| Text Generation (AR) | $x$: sequence with mask tokens, $y$: unmasked sequence |

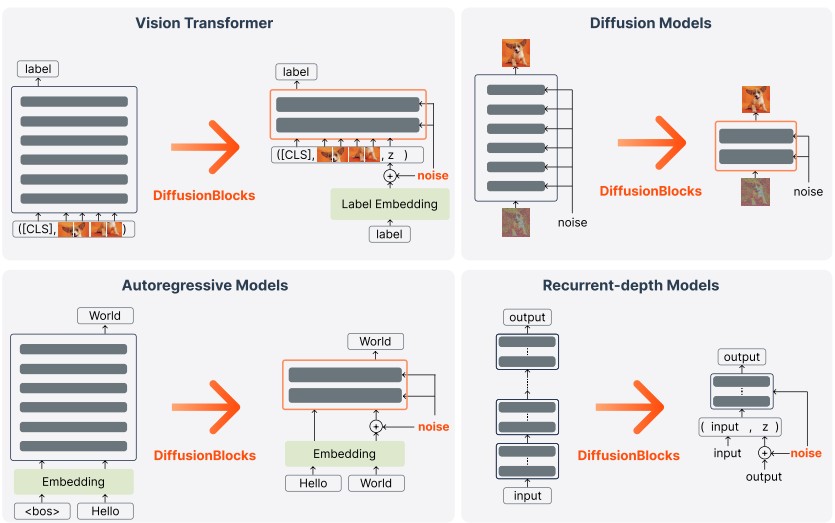

Figure 5: **Converting different architectures to DiffusionBlocks: Training.** During training, noise is added to target outputs (labels, embeddings, or images) and each block learns to denoise within its assigned noise range. Blocks are sampled randomly and trained independently, requiring gradients for only one block at a time.

embeddings $\mathbf{x}$, and the noisy label embedding $\mathbf{z}_\sigma$, where $\mathbf{z}_\sigma = \mathbf{y}_{\text{emb}} + \sigma\epsilon$ and $\mathbf{y}_{\text{emb}} \in \mathbb{R}$ is the learnable continuous embeddings for the class label $y$. Each block $b$ learns to denoise this label representation conditioned on the patch embeddings $\mathbf{x}$. The training loss is the standard cross-entropy between the classification head's output logits (applied to the [CLS] token) and the true class labels, following the conventional ViT training procedure.

For diffusion models (top right), DiffusionBlocks provides a natural fit: these models already operate by denoising, so partitioning simply assigns different noise ranges to different blocks without architectural modifications. The standard denoiser $D_{\boldsymbol{\theta}}(\mathbf{z}_\sigma, \sigma)$ becomes $D_{\boldsymbol{\theta}_b}(\mathbf{z}_\sigma, \sigma)$ for block $b$.

For discrete output spaces like language modeling (bottom left), we operate in the embedding space following prior works (Dieleman et al., 2022; Li et al., 2022; Gulrajani & Hashimoto, 2023; Lovelace et al., 2023). Noise is added after the embedding layer: given input tokens $\mathbf{x}$, we compute $\mathbf{z} = f_{\text{in}}(\mathbf{x})$, then add noise $\mathbf{z}_\sigma = \mathbf{z} + \sigma\epsilon$. For autoregressive models, the denoiser $D_{\boldsymbol{\theta}_b}(\mathbf{z}_{i,\sigma}, \mathbf{z}_{<i}, \sigma)$ recovers the clean embedding of token $i$ from its noisy version, conditioned on previous clean token embeddings $\mathbf{z}_{<i}$. We minimize cross-entropy loss instead of L2 loss.

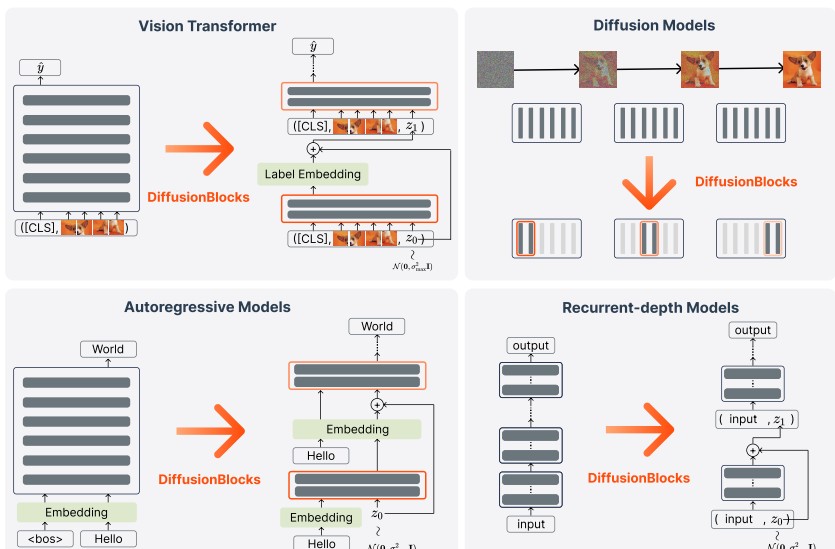

Figure 6: **Converting different architectures to DiffusionBlocks: Inference.** During inference, blocks are applied sequentially from $\sigma_{\max}$ to $\sigma_{\min}$. The figure shows the first denoising step where block $b = 1$ transforms pure noise $\mathbf{z}_0$ into the next state $\mathbf{z}_1$. Only the relevant block is active at each noise level, providing memory efficiency. $\oplus$ denotes the Euler step in Eq. (5).

For recurrent-depth architectures that apply the same network $K$ times (bottom right), we interpret the entire recurrence as a diffusion process. Instead of training with $K$ forward passes through recurrent iterations, we train the network as a denoiser $D_\theta(\mathbf{z}_\sigma, \mathbf{x}, \sigma)$ by sampling $\sigma \sim p_\sigma$ and performing a single forward pass to map noisy input to clean output, reducing computational cost by factor $K$ while maintaining the original $K$-iteration inference procedure.

Beyond these adaptations, DiffusionBlocks also applies to diffusion language models (Austin et al., 2021; Lou et al., 2024; Sahoo et al., 2024; Shi et al., 2024), where the framework provides additional benefits for text generation. We provide a detailed treatment of this application in Appendix D. These diverse applications demonstrate that DiffusionBlocks provides a general recipe for transforming various architectures into memory-efficient, independently trainable components.

## C   IMPLEMENTATION DETAILS IN DIFFUSIONBLOCKS

We introduce several practical considerations for effective training and inference.

**Overlap between blocks.**   To smooth transitions across block boundaries, we slightly extend each block's noise interval in log-$\sigma$ space. For a block $b$ responsible for $[\sigma_b, \sigma_{b-1}]$ with $\sigma_{b-1} > \sigma_b$, we define $\alpha_b := (\sigma_{b-1}/\sigma_b)^\gamma$, where $\gamma \geq 0$, and train over the expanded range $[\sigma_b/\alpha_b, \alpha_b\sigma_{b-1}]$. Here $\gamma$ controls the degree of overlap: $\gamma=0$ recovers non-overlapping intervals, while $\gamma > 0$ yields smoother transitions between blocks. In practice, we found $\gamma \in [0.0, 0.1]$ effective, and we use $0.05$ by default and $0.1$ for text generation.

**Weighting and preconditioning.**   Following the EDM framework (Karras et al., 2022), we use the weighting function: $w(\sigma) = (\sigma^2 + \sigma_{\text{data}}^2)/(\sigma \cdot \sigma_{\text{data}})^2$ where $\sigma_{\text{data}} = 0.5$ for all experiments. The weighting is crucial for equi-probability partitioning to work effectively, as it counteracts the sampling bias introduced by the log-normal distribution $p_\sigma$. We also adopt EDM's preconditioning scheme, which involves input scaling to ensure stable training dynamics across all noise levels. See Karras et al. (2022) for more details.

**Normalizing embeddings.**   For tasks where the target variables are discrete (e.g. class labels in image classification or token ids in text generation), DiffusionBlocks operates the diffusion process

in the continuous embedding space (see Appendix B). A known issue in continuous relaxation of discrete variables is *embedding collapse*, where all learned embeddings correspond to the same vector (Dieleman et al., 2022). To prevent this, we follow the regularization strategy introduced Dieleman et al. (2022) and apply L2 normalization to the embeddings.

**Training and inference details.** For training efficiency, blocks are randomly sampled per iteration, requiring memory for only $L/B$ layers. Blocks can alternatively be trained in parallel across multiple GPUs when available. During inference, we generate samples by sequentially applying blocks from $\sigma_{\max}$ to $\sigma_{\min}$. While we use Euler steps in our experiments due to the natural correspondence between residual connections and Euler discretization (Section 2.2), our framework is not limited to this choice. By modifying the inter-block connections to match the discretization scheme of other solvers, any diffusion sampling methods (Song et al., 2021a; Lu et al., 2023; Zhao et al., 2023) can be employed. We leave this exploration for future work.

# D    MASKED DIFFUSION LANGUAGE MODELS AS DIFFUSIONBLOCKS

## D.1    CONTINUOUS-TIME FORMULATION

We first recall the continuous-time formulation of masked diffusion language models (Sahoo et al., 2024; Shi et al., 2024). Let $\mathbf{x}_0 = (x_{01}, \ldots, x_{0n})$ denote a sequence of tokens and let $\alpha(t) : [0, 1] \to [1, 0]$ denote the masking schedule at continuous time $t \in [0, 1]$, where $\alpha(t)$ represents the probability of remaining unmasked. The forward process progressively masks tokens as:

$$q(\mathbf{x}_t \mid \mathbf{x}_0) = \prod_{i=1}^{n} q(x_{ti} \mid x_{0i}) \quad \text{where} \quad x_{ti} = \begin{cases} x_{0i}, & \text{with prob. } \alpha(t), \\ [\text{MASK}], & \text{with prob. } 1 - \alpha(t). \end{cases} \tag{7}$$

The training objective in continuous form is:

$$\mathcal{L}(\boldsymbol{\theta}) = \mathbb{E}_{\mathbf{x}_0} \int_0^1 \frac{-\alpha'(t)}{1 - \alpha(t)} \mathbb{E}_{\mathbf{x}_t \sim q(\mathbf{x}_t \mid \mathbf{x}_0)} \left[ \sum_{i : x_{ti} = [\text{MASK}]} \text{CE}(f_\theta(\mathbf{x}_t, t)_i, x_{0i}) \right] dt, \tag{8}$$

where $\alpha'(t) = d\alpha/dt < 0$ and CE denotes cross-entropy loss. This form is equivalent to the continuous-time NELBO (Shi et al., 2024; Sahoo et al., 2024), but expressed with a nonnegative weight multiplying CE, which avoids sign ambiguity.

## D.2    PARTITIONING INTO DIFFUSIONBLOCKS

To enable block-wise training, we partition the objective in Eq. (8) into $B$ disjoint intervals in $t$. The expected number of masked positions at time $t$ is $n(1 - \alpha(t))$, so the effective density of contributions is

$$\frac{-\alpha'(t)}{1 - \alpha(t)} \cdot (1 - \alpha(t)) = -\alpha'(t). \tag{9}$$

Hence, the contribution of interval $[t_a, t_b]$ is

$$\int_{t_a}^{t_b} -\alpha'(t) \, dt = \alpha(t_a) - \alpha(t_b). \tag{10}$$

This shows that the training mass is distributed uniformly in $\alpha$, not in $t$.

Therefore, the natural partition boundaries are defined by equal decrements of $\alpha$:

$$\alpha_b = 1 - \tfrac{b}{B}, \quad b = 0, \ldots, B, \tag{11}$$

with corresponding time boundaries obtained by inversion:

$$t_b = \alpha^{-1}\left(1 - \tfrac{b}{B}\right). \tag{12}$$

For a linear schedule $\alpha(t) = 1 - t$, this simply yields $t_b = b/B$. Each block $b$ is then trained independently on its assigned interval:

$$\mathcal{L}_b(\boldsymbol{\theta}_b) = \mathbb{E}_{\mathbf{x}_0} \int_{t_{b-1}}^{t_b} \frac{-\alpha'(t)}{1 - \alpha(t)} \mathbb{E}_{\mathbf{x}_t \sim q(\mathbf{x}_t \mid \mathbf{x}_0)} \left[ \sum_{i : x_{ti} = [\text{MASK}]} \text{CE}\left(D_{\boldsymbol{\theta}_b}(\mathbf{x}_t, t)_i, x_{0i}\right) \right] dt, \tag{13}$$

where $D_{\boldsymbol{\theta}_b}$ denotes the denoiser assigned to block $b$. The global loss decomposes as $\mathcal{L} = \sum_{b=1}^{B} \mathcal{L}_b$.

This derivation shows that DiffusionBlocks in masked diffusion models amounts to partitioning the masking schedule $\alpha(t)$ rather than time. Each block is responsible for an equal decrement in $\alpha(t)$, i.e. an equal share of the total "demasking work", which ensures balanced parameter utilization and true independence across blocks. This construction is directly analogous to the equi-probability partitioning in continuous diffusion models described in Section 3.3.

## E    EXPERIMENTAL DETAILS

Unless otherwise specified, all experiments use the following settings. For DiffusionBlocks, we adopt the EDM framework (Karras et al., 2022) with default parameters: log-normal noise distribution with $P_{\text{mean}} = -1.2$ and $P_{\text{std}} = 1.2$, noise range $[\sigma_{\min}, \sigma_{\max}] = [0.002, 80]$, and preconditioning following the recommended configuration. Inference uses Euler sampling with 50 steps unless stated otherwise. During training, blocks are sampled uniformly at random for each iteration.

### E.1    VISION TRANSFORMERS FOR IMAGE CLASSIFICATION

For image classification experiments in Section 5.1, we use a 12-layer ViT with patch size 4, 128 hidden dimensions, 4 attention heads, and 0.1 dropout, partitioned into $B$=3 blocks (4 layers each). We train for 500 epochs with batch size 128 and AdamW optimizer with learning rate $5 \times 10^{-4}$. We employ a cosine learning rate scheduler with a 10-epoch linear warmup. As data augmentation, we apply random horizontal flipping ($p = 0.5$) and *RandAugment* (Cubuk et al., 2020) as data augmentation.

Figure 5 (top left) illustrates the DiffusionBlocks adaptation for ViT. We add noise to the class label embeddings and concatenate them with the patch embeddings. Each block learns to denoise the label embedding conditioned on the patch embeddings. We use an overlap ratio $\gamma = 0.05$ and perform 4 denoising steps during inference (matching $L/B$= 12/3). The classification head is applied after the final denoising step to produce class predictions. We minimize cross-entropy loss between predicted and true class labels during training. For the Forward-Forward baseline, we adapt the Contrastive Forward-Forward (FF) (Aghagolzadeh & Ezoji, 2025) implementation to ViT [3].

### E.2    DIFFUSION MODELS FOR IMAGE GENERATION

For image generation experiments in Section 5.2, we use DiT-S/2 (12 layers) for CIFAR-10 and DiT-L/2 (24 layers) for ImageNet-256. Both models are partitioned into $B = 3$ blocks. Training follows the EDM framework with classifier-free guidance (Ho & Salimans, 2021) (10% label dropout). For CIFAR-10, we train for 100 epochs with batch size 512 and AdamW optimizer with learning rate $10^{-4}$. For ImageNet, we resize to 256×256 and encode images by a pre-trained VAE (Peebles & Xie, 2023)[4]. We also train 100 epochs with batch size 512 and AdamW optimizer with learning rate $5 \times 10^{-5}$. Overlap ratio is set to $\gamma = 0.05$.

In evaluation, we apply Euler sampling with 50 steps and classifier-free guidance (scale 2.0) on both CIFAR-10 and ImageNet experiments. FID is computed using 50,000 generated samples against the training and test sets, with the minimum of three evaluations reported following Karras et al. (2022). For the training set, we use the official ADM (Dhariwal & Nichol, 2021) evaluation suite, which computes FID against the entire training set as the reference distribution. For the test split, we compute FID using `clean-fid` (Parmar et al., 2022).

### E.3    MASKED DIFFUSION MODELS FOR TEXT GENERATION

In Section 5.3, we follow MD4's training protocol with 256 sequence length, AdamW optimizer with learning rate $3 \times 10^{-4}$, weight decay 0.03, and 2,000 linear warmup steps. Training runs for 100 epochs with batch size 256. The 12-layer DiT-based transformer (Lou et al., 2024; Sahoo et al., 2024) uses 768 hidden dimensions and 12 attention heads, partitioned into $B$=3 blocks with

---

[3] https://github.com/HosseinAghagol/ContrastiveFF
[4] stabilityai/sd-vae-ft-ema

overlap ratio $\gamma = 0.05$. Masking schedule follows MD4's linear schedule. For block partitioning in discrete diffusion, we apply equi-probability partitioning to the masking ratio distribution rather than continuous noise levels in Appendix D. Bits-per-character (BPC) is evaluated on the text8 test set following Shi et al. (2024).

### E.4 AUTOREGRESSIVE MODELS FOR TEXT GENERATION

In Section 5.4, we use a 12-layer Llama-2-style transformer (Touvron et al., 2023) augmented with time conditioning as in DiT (Peebles & Xie, 2023) with 768 hidden dimensions, 12 attention heads, and the Llama-2 tokenizer with 32K vocabulary size. The model is partitioned into $B$=4 blocks with an overlap ratio $\gamma$=0.1. Training uses sequence length 256 for LM1B and 3072 for OWT, batch size 256, AdamW with learning rate $3 \times 10^{-4}$, and 2500 warmup steps for 10 epochs.

Since DiffusionBlocks is not derived from ELBO-based objectives, computing traditional perplexity is non-trivial. Instead, we evaluate using MAUVE scores following SEDD (Lou et al., 2024), which measures the similarity between generated and real text distributions. For each test sample, we generate 5 continuations of 50 tokens from 1K prompts and compute MAUVE against 1K reference samples with the scaling factor 0.2. Additionally, we report generative perplexity, commonly used in diffusion language models (Lou et al., 2024; Sahoo et al., 2024), by computing the perplexity of generated text using teacher models (`Llama-2-7B`[5] and `GPT2-XL` (Radford et al., 2019)[6]). For generations, we use top-p sampling (0.95) for the baseline and 4 diffusion steps with greedy sampling for DiffusionBlocks. The OWT test set is created by splitting 10% of the data since no official test set exists.

Applying DiffusionBlocks to autoregressive models requires maintaining causal consistency during training. When denoising future tokens, the model must condition on clean past tokens rather than noisy ones to preserve the autoregressive property. Following Block Diffusion (Arriola et al., 2025), we implement this using sequence concatenation: noisy and clean sequences are concatenated with a modified causal attention mask that allows noisy tokens to attend to their corresponding clean past tokens while preventing information leakage. This approach doubles sequence memory but maintains single forward pass efficiency. An alternative implementation computes key-value pairs separately for clean and noisy sequences, combining them during attention computation. This requires two forward passes but uses standard sequence memory. We adopt the concatenation approach for computational efficiency.

### E.5 RECURRENT-DEPTH MODELS

For `Huginn` (Geiping et al., 2025) described in Section 5.5, we use the default configuration: 2 prelude layers, 4-layer recurrent block, and 2 coda layers following `Pythia-70M` (Biderman et al., 2023)[7] architecture with 512 hidden dimensions and 8 attention heads. Unlike other architectures, recurrent-depth models do not require block partitioning since the entire network is applied recurrently. Instead, we train the full network as a denoiser by sampling different noise levels $\sigma$ at each training step. While baseline `Huginn` uses stochastic recurrence depth (average 32 iterations) with truncated BPTT (8 steps), DiffusionBlocks trains with single-pass diffusion. We train on LM1B for 15 epochs compared to `Huginn`'s 5 epochs. Despite this, our approach uses approximately $10\times$ less total computation since we avoid the $32\times$ recurrent iterations during training.

### E.6 ABLATION STUDIES

#### E.6.1 COMPARISON WITH NOPROP

We follow the experimental protocol of NoProp (Li et al., 2025). In the absence of publicly available code, we implemented their `NoProp-DT` architecture augmented with time conditioning from `NoProp-CT`, following their specifications (Figure 5 in their paper). Training follows `NoProp-CT`'s hyperparameters with AdamW optimizer, learning rate $10^{-4}$, batch size 128, and 1000 epochs on CIFAR-100. For DiffusionBlocks, we use $B$=3 blocks with overlap ratio $\gamma = 0.1$.

---

[5] `https://huggingface.co/meta-llama/Llama-2-7b-hf`
[6] `https://huggingface.co/openai-community/gpt2-xl`
[7] `https://huggingface.co/EleutherAI/pythia-70m`

Table 9: **ViT results on Tiny-ImageNet.** DiffusionBlocks shows consistent performance on intermediate-scale classification dataset.

| Method | Accuracy ($\uparrow$) |
|---|---|
| ViT | 35.32 |
| **+ DiffusionBlocks** | **36.16** |

Following `NoProp-CT`'s evaluation protocol, we use 1000 Euler sampling steps instead of our default 50.

We attempted to adapt Forward-Forward (FF) algorithm (Hinton, 2022) as an additional baseline to NoProp's architecture for Table 6. However, without publicly available code and with no specified adaptation procedure, the implementation requires numerous design decisions. Our attempts achieved only 1% accuracy, highlighting the fundamental incompatibility: NoProp's architecture is specifically designed for their method (type (e) in their Figure 2), while FF requires contrastive positive/negative samples (type (d)). Successfully bridging these paradigms may require innovations beyond straightforward adaptation. This highlights a key distinction between approaches. NoProp does not provide guidance for adapting to other methods or architectures. DiffusionBlocks instead offers a systematic procedure for converting existing Transformer-based networks into block-wise trainable models. This recipe enabled successful application to modern architectures with minimal modifications, demonstrating the generality of our framework.

### E.6.2 Design choice analysis

All ablation studies follow the configurations described in Appendix E.2. We report FID scores on the test splits. For partitioning experiments, we test both uniform partitioning (equal intervals in log-space) and our equi-probability method. Layer distribution indicates the number of layers in each block. For block count experiments, we vary $B$ from 2 to 6 while keeping total layers fixed at 12. We disabled the block overlap ($\gamma = 0.0$) in Section C to isolate the effectiveness of each component.

## F Additional experiments

### F.1 Image Classification Experiment on Tiny ImageNet

To further evaluate the effectiveness of DiffusionBlocks on classification tasks beyond CIFAR-100, we conducted an additional experiment on the Tiny ImageNet dataset (Le & Yang, 2015). This dataset consists of 200 classes, 100,000 training images with each image resized to $64\times64$ resolution. Tiny-ImageNet provides a more challenging and higher-resolution benchmark than CIFAR-100.

We trained a 12-layer Vision Transformer (ViT) with patch size 4, hidden size 768, and 12 attention heads. Both the baseline ViT and DiffusionBlocks models were trained for 100 epochs using a batch size of 256 and the AdamW optimizer with a learning rate of $10^{-4}$. For DiffusionBlocks, we used $B = 2$ blocks (each containing 6 layers).

Table 9 demonstrates that DiffusionBlocks maintains competitive performance relative to the baseline ViT, consistent with our findings on CIFAR-100 as well as our large-scale classification experiments in language modeling (LM1B and OpenWebText in Table 3, 4, 5). These results further indicate that DiffusionBlocks remains effective as a classifier across different data modalities, resolutions, and dataset scales.

### F.2 Effect of block count on CIFAR-10

To examine whether the design trends observed on ImageNet in Table 8 generalize to different datasets, we additionally evaluate the effect of the number of blocks on CIFAR-10. This experiment allows us to assess whether the behavior of DiffusionBlocks remains consistent across datasets of different scales and complexities. We use the same DiT-S/2 architecture described in Section 5.2,

Table 10: **Effect of block count on CIFAR-10.** Moderate block counts (2–3) achieve the best FID, showing consistent trends with ImageNet in Table 8.

| Number of Blocks | FID ($\downarrow$) | L/B ($\downarrow$) | Relative Speed |
|---|---|---|---|
| $B = 1$ | 39.83 | 12 | $1.0\times$ |
| $B = 2$ | **35.47** | 6 | $2.0\times$ |
| $B = 3$ | **38.03** | 4 | $3.0\times$ |
| $B = 4$ | 45.43 | 3 | $4.0\times$ |
| $B = 6$ | 53.32 | 2 | $6.0\times$ |

Table 11: **Effect of block count on text generation (LM1B).** Best performance is achieved with $B$=4.

| Number of Blocks | MAUVE ($\uparrow$) | Layers per Block ($\downarrow$) | Relative Speed |
|---|---|---|---|
| $B = 2$ | 0.61 | 6 | $2.0\times$ |
| $B = 3$ | 0.65 | 4 | $3.0\times$ |
| $B = 4$ | **0.67** | 3 | $4.0\times$ |
| $B = 6$ | 0.62 | 2 | $6.0\times$ |

training under the EDM framework while varying the number of blocks $B \in \{1, 2, 3, 4, 6\}$ and disabling block overlap ($\gamma = 0.0$) to isolate the effectivenss of the number of blocks $B$.

As shown in Table 10, smaller block counts tend to achieve better FID scores, and $B = 2$ or $B = 3$ provides strong performance. This trend matches the observations in Table 8. These results indicate that the effectiveness of using a moderate number of blocks is consistent across datasets of varying scale, supporting the validity of the design choices analyzed in Section 5.6.

### F.3 EFFECT OF BLOCK COUNT ON TEXT GENERATION

Table 11 shows the effect of varying the number of blocks for autoregressive language modeling on LM1B with overlap ratio $\gamma = 0.0$.

The optimal number of blocks differs between tasks: image generation achieves best FID with $B$=2 or $B$=3 (Table 8), while language modeling achieves best MAUVE with $B$=4. This motivated our choice of $B$=4 for language modeling experiments in the main paper.

## G COMPARISON WITH ACTIVATION CHECKPOINTING

DiffusionBlocks and activation checkpointing (also known as activation recomputation, gradient checkpointing, or rematerialization) offer fundamentally different trade-offs and can be powerfully combined.

The key distinction lies in what each method reduces. Activation checkpointing reduces only activation memory, leaving parameters, gradients, and optimizer states unchanged. In contrast, DiffusionBlocks reduces all memory components by a factor of $B$. This distinction becomes increasingly critical as modern models grow larger.

To illustrate this difference, consider an $L$-layer network where each layer has parameter size $P$ and activation size $A$. With Adam optimizer (requiring $2P$ for momentum and variance), each layer needs $4P$ memory for parameters, gradients, and optimizer states. Standard training thus requires $(4P + A)L$ total memory. Activation checkpointing reduces this to $4PL + A$ by rematerializing activations only when needed (though this is an optimistic estimate that ignores the memory cost of the checkpoints). DiffusionBlocks, by training $B$ independent blocks, requires $(4P + A)(L/B)$. Since $L > B$, combining DiffusionBlocks and activation checkpointing uses the least memory among these four patterns.

Regarding computational costs, it is empirically known that activation checkpointing increases the training time by a factor of approximately 4/3, and this holds true when combined with the proposed method. This is justified as follows. With a forward pass computation cost of $F$, a backward pass requires approximately $2F$ (computing Jacobians and weight gradients). Standard training uses $3F$ cost per iteration, while activation checkpointing increases this to $4F$ due to recomputation. DiffusionBlocks maintains this ratio when combined with checkpointing.

Beyond memory reduction, DiffusionBlocks offers unique advantages regarding training time: each block can be trained in an embarrassingly parallel manner. This means each block can be trained in parallel with absolutely no communication overhead. This provides an additional advantage over activation checkpointing, especially when computational resources are abundant.

Table 12: Wall-time comparison on ViT. The aggregated DiffusionBlocks time is computed by multiplying the measured per-block iteration time by $B = 3$.

| Method | Wall time (sec/iter) |
| --- | --- |
| ViT | 0.0507 |
| DiffusionBlocks: per-block time (4 layers) | 0.0181 |
| DiffusionBlocks: aggregated time ($0.0181 \times 3$) | 0.0543 |

## H    TRAINING AND INFERENCE EFFICIENCY

This section provides a detailed analysis of the computational efficiency and wall-time characteristics of DiffusionBlocks.

**Training efficiency.**    Consider an $L$-layer network trained for $K$ iterations. Standard end-to-end backpropagation performs $K \times L$ layer evaluations. DiffusionBlocks trains only $L/B$ layers at a time; training all $B$ blocks for $K$ iterations each performs $(L/B) \times B \times K = L \times K$ layer evaluations. Thus, DiffusionBlocks requires the *same* total amount of computation as standard training, while reducing memory usage by a factor of $B$.

To validate this theoretical equivalence, we measured the per-iteration wall time using a 12-layer ViT on a single H100 80GB GPU, averaging over 100 iterations. As summarized in Table 12, standard training requires 0.0507 seconds per iteration for all 12 layers. Under DiffusionBlocks with $B = 3$, each block (4 layers) takes 0.0181 seconds per iteration (measured). The total per-iteration wall time for DiffusionBlocks is therefore obtained by summing the independently trained blocks, computed as $0.0181 \times 3 = 0.0543$ seconds. The resulting end-to-end wall time is thus comparable to standard training, with the small difference attributable to the noise-level conditioning introduced during the DiffusionBlocks conversion (Section 3.1).

**Inference efficiency.**    For inference, we ensure that the total amount of computation matches that of the baseline model. For a 12-layer network, the baseline performs a single forward pass through all 12 layers. Under DiffusionBlocks with $B = 3$, we perform three denoising steps, each invoking the corresponding 4-layer block once. The total compute therefore corresponds to the same 12 layer evaluations as in standard inference.

For diffusion models used in image generation, the computational benefit is even more pronounced. Standard diffusion models must apply the full network for every denoising step. With 50 denoising steps, a 12-layer DiT requires $12 \times 50$ layer evaluations. In DiffusionBlocks, each denoising step applies only the block responsible for that noise level, which contains 4 layers when $B = 3$. This reduces the total compute to $4 \times 50$, achieving a $B$-fold reduction in inference cost. The 50 denoising steps are assigned to blocks according to the equi-probability partitioning in Section 3.3, so that each block is used approximately the same number of times during inference. Euler sampling is used for simplicity, and, as shown in Section 2.2, it is computationally equivalent to a residual update, requiring no additional overhead.

