# OpenReview forum: "DiffusionBlocks: Block-wise Neural Network Training via Diffusion Interpretation"
_ICLR.cc/2026/Conference — ICLR 2026 Poster_

### Official Review · Reviewer_ZJo4 · 2025-10-30

**Soundness:** 2
**Presentation:** 3
**Contribution:** 3
**Rating:** 6
**Confidence:** 4

**Summary:**

This paper presents a novel method to transform modern networks with residual connections (e.g., ViT) to independent trainable blocks that learns a diffusion denoising process for a predefined noise range. During training, all blocks rely only on the input x and not output of previous/other blocks. During inference, input x is passed as a condition to blocks starting with the largest noise range to the smallest noise range. The method does not require backpropagation across all blocks, reducing memory usage. Experiments on numerous tasks demonstrate promising results.

**Strengths:**

- The main idea of this paper is novel (concurrent to NoProp). It tries to convert the foward propagation proccess of residual-based architectures to a diffusion denoising process.
- The method is evaluated on numerous tasks (image + text classifcation and generation)
- There are some theory analysis of the partitioning approach

**Weaknesses:**

- In Table 2, the number of training epoch/iterations are not reported. For DiT-L/2 on ImageNet256x256, what are the FID for with and without classifier free guidance (CFG). The current result does not state whether CFG is used.
- To my understanding Equation 4 should only denote one residual connection. In ViT, one residual connection is for the self-attention operation and one residual connection is for the MLP. To be rigourous with the thoery, each denoising block should denote a single residual connection so each ViT block should be decomposed into two diffusion blocks. In the current implementation ViT/DiT with 12/24 layers are partitioned into 3 blocks. From this perspective, I don't think the thoery fully back up the proposed partition approach.
- Despite that the proposed method uses less memory than standard models, diffusion processes are more complex. I am worried that the acutal speed of the method is much slower than standard ViT. The authors should report real time comparison of training and inference speeds on GPUs for example on the image classifcation and generation tasks.
- The authors should report the number of denoising steps for each block during inference.

**Questions:**

- Each block learns difference noise levels independently. In this case, why use mulltiple blocks and not just one large block to learn all noise ranges?
- The last block of the network reconstruct the original input. Why can this be used for classification? Conditioned on the input, the denoising blocks reconstruct the input. This does not yield a standard representation like other models that can be used for deterministic tasks. The authors have demonstrated classifcation performance on CIFAR but I think this is not sufficient. I would really like to see whether the approach works for ImageNet-1k as well as downstream dense prediction tasks such as semantic segmentation. Due to the time limit of the rebuttal, partial results are ok, for example, training only 100 epochs instead of 300 (standard ImageNet-1k training setting).
- If I want to finetune a pretrained model on a dataset, can the weights and knowledge learned by the pretrained model ViT be used with this approach?

Overall, I think the idea is novel, but there are some questions that needs clarifying and I'm have concerns on the practical application and performance of the proposed approach.

---

> ### Author Response · Authors · 2025-11-20
>
> We greatly appreciate your careful examination of our experimental methodology and insightful recognition of DiffusionBlocks' broader potential, particularly your forward-looking perspective on future research directions. For clarity, all modifications made in response to your comments are highlighted in $\textcolor{magenta}{magenta}$ in the revised manuscript.
>
> ### 1. Training Configuration for Image Generation
> > In Table 2, the number of training epoch/iterations are not reported. For DiT-L/2 on ImageNet256x256, what are the FID for with and without classifier free guidance (CFG). The current result does not state whether CFG is used.
>
> Thank you for highlighting the importance of experimental transparency. While complete training details, including the number of epochs (100) and batch size (512), are documented in Appendix D.2, we agree that key configuration choices should be stated more prominently.
>
> Regarding classifier-free guidance (CFG), all results in Table 2 employ CFG with scale 2.0, and we use 10% label dropout during training to enable effective conditional generation. Since this is essential for reproducibility, we have updated Appendix D.2 to explicitly include this configuration.
>
> A noteworthy strength of DiffusionBlocks is that it integrates naturally with advances in diffusion models. Although we intentionally disabled CFG for classification and text generation tasks to ensure a fair comparison with baselines that do not natively support CFG, DiffusionBlocks itself fully supports CFG and other modern diffusion techniques such as fast samplers across all architectures. This capability suggests promising opportunities for extending diffusion-based enhancements to domains where such techniques were previously unavailable.
>
> ### 2. Correspondence between ViT layers and denoising steps
> > To be rigorous with the theory, each denoising block should denote a single residual connection so each ViT block should be decomposed into two diffusion blocks.
>
> We would like to note that our scheme of converting residual networks into Diffusion blocks leaves much freedom to the user.
> In particular, neither our theoretical claims nor the method necessitates the usage of all the original skip connections present in the original networks for conversion.
>
> For example, when $x_1 =  g_1(x_0) + x_0$  and $x_2 =  g_2(x_1) + x_1$ where $g_1, g_2$ are some dimension-preserving module, then we can allow:
> $f_{\theta_1}(x_0) =  g_1(x_0) + x_0$,
> $f_{\theta_2}(x_1) =  g_2(x_1) + x_1$.
> And let $f_{\theta_b} = f_{\theta_2} \circ f_{\theta_1}$ constitute the core of “a” block $b$ in Section 3.1.
> Note that in this example, we will not reuse the skip connections in $f_{\theta_1}, f_{\theta_2}$ in carrying out steps 1, 2, and 3, i.e., these skip connections will remain **internal** in the Euler update of Eq. (5).
>
> For our conversion of ViT, we take the same approach.
> Our formulation treats **each complete ViT block as a single Euler step** in Eq. (4), rather than treating the self-attention and MLP residual paths as two independent discretization steps.
> As we illustrate in Figure 6 in Appendix B,  the Euler updates then occur **between blocks**, and the residual paths inside a ViT layer remain **internal** to the denoiser associated with that block.
>
> We emphasize that this is a “design choice” that we deliberately leave to the user in our *diffusion-blockification*.   As you point out, we can treat each each architectural component with a skip connection as *an instance of a block* and the re-use the skip connection as a part of Eq. (5).  Similarly to the choice of the stepsize in Euler updates, this freedom does not undermine our theoretical soundness and its efficacy depends on the balance of the chosen network and task complexity.
> This is the very essence of the block partitioning strategy we mentioned in Section 3.3.
>
> We do note that, as we ablated in Table 7, 8, and 9, the decomposition of the original networks into finest component did not yield the optimal results, and finding the optimal block partitions for a given task is an important future work.
> We have added this important consideration about the granularity of block partitioning in Section 6.
>
> ---
>
> Due to the character limit, we will address the remaining points in our second response.

---

> > ### Author Response · Authors · 2025-11-20
> >
> > ### 3. Training and inference efficiency
> > > Despite that the proposed method uses less memory than standard models, diffusion processes are more complex. I am worried that the acutal speed of the method is much slower than standard ViT. The authors should report real time comparison of training and inference speeds on GPUs for example on the image classifcation and generation tasks.
> >
> > > The authors should report the number of denoising steps for each block during inference.
> >
> > Thank you very much for raising these practical concerns. While diffusion models are indeed built on a more complex theoretical formulation, in practice the actual training and inference time of DiffusionBlocks is almost the same as that of the baseline models.
> >
> > Regarding the training time, we ensured a fair comparison by matching the total number of training iterations between the baseline and DiffusionBlocks. When the network is divided $L$-layer network into $B$ blocks, DiffusionBlocks trains $B$ smaller networks, each of which has $L/B$ layers. If we keep the number of iterations unchanged, the cost of training each network is roughly $1/B$, and training all $B$ networks results in a total cost that is essentially the same as the baseline. Although the inputs, outputs, and losses differ between the two methods, the dominant cost still comes from the forward and backward computations.
> >
> > To verify this empirically, we measured the average wall time over 100 iterations for a 12 layer ViT. The results are as follows.
> >
> > - Standard training (12 layers): 0.0507 [sec/iter]
> > - DiffusionBlocks (B = 3):
> >   - Per block (4 layers): 0.0181 [sec/iter]
> >   - Total over all blocks: $0.0181×3=0.0543$ [sec/iter]
> >
> > The small overhead of about 7 percent comes from the time conditioning. Furthermore, when considering parallelization, DiffusionBlocks is advantageous in practice because the $B$ block trainings are embarrassingly parallel, meaning that each block can be trained independently without any communication overhead.
> >
> > The inference time follows the same principle. For example, in the case of a 12-layer model, the standard baseline performs one forward pass through all 12 layers. In DiffusionBlocks with B=3, we set the number of denoising steps to 3 and apply each block (4 layers) once. The total compute therefore corresponds to 12 layers, which matches the baseline. For simplicity, we use Euler sampling, and as discussed in Section 2.2, this operation is equivalent to the residual update and does not introduce any additional computational complexity.
> > For image generation with diffusion models, the situation is even more favorable. The baseline requires running the full network for all 50 denoising steps, so a 12 layer model performs $12 \times 50$ layer forward computations. In DiffusionBlocks, one denoising step corresponds to applying only the block assigned to that noise level, which has 4 layers. This results in $4 \times 50$ layer forward computations, achieving a factor of $B$ speedup. In this case, the 50 steps are divided among the blocks according to the block partitioning strategy in Section 3.3, so that each block is used approximately the same number of times.
> >
> > We have added these clarifications to the revised manuscript in Appendix H.
> >
> > ----
> >
> > Due to the character limit, we will address the remaining points in our third response.

---

> > > ### Author Response · Authors · 2025-11-20
> > >
> > > ### 4. Multiple blocks instead of one block to learn all noise
> > > > Each block learns difference noise levels independently. In this case, why use mulltiple blocks and not just one large block to learn all noise ranges?
> > >
> > >
> > > Thank you for raising this fundamental question.
> > > Using a single block to learn all noise levels is indeed equivalent to the standard diffusion model design in image generation, where one denoiser network handles the entire noise range. In Table 2, this corresponds to the baseline that covers all noise levels with a single network. DiffusionBlocks and the single-block baseline have the same total number of parameters, but DiffusionBlocks provides two key advantages: (1) Block-wise training with B-fold memory reduction (2) B-fold inference speedup as discussed in 3.
> > > At every denoising step, a single-block model must forward-pass the entire network, while our approach  forward-pass only one of the B blocks. Please see  the top right panel of Figure 6 for the illustration of this difference.
> > >
> > > These benefits arise from our partitioning strategy (Section 3.3), which allocates noise intervals in accordance to their intrinsic difficulty. This allows the model’s parameters to be used more efficiently than when a single network must cover the entire noise spectrum.
> > >
> > > Beyond image generation, the idea of training a single network to learn all information is conceptually related to Neural ODEs [2] and recurrent-depth models [3][4], which repeatedly apply the same network to update their states. Although these methods do not use diffusion, they share in common the strategy of forward passing a single, heavy network multiple times.
> > >
> > > In the light of the empirical observations that our method does not compromise the performance,  we believe we can say that our strategy of “independently training multiple blocks” is computationally more efficient than the classical strategy.
> > >
> > > ----
> > >
> > > [2] Chen et al., Neural Ordinary Differential Equations, 2018.
> > >
> > > [3] Dehghani et al., Universal Transformers, 2018.
> > >
> > > [4] Geiping et al., Scaling up Test-Time Compute with Latent Reasoning: A Recurrent Depth Approach, 2025.
> > >
> > > ----
> > >
> > > Due to the character limit, we will address the remaining points in our fourth response.

---

> ### Author Response · Authors · 2025-11-20
>
> ### 5. On the use of DiffusionBlocks for classification tasks
> > The last block of the network reconstruct the original input. Why can this be used for classification? Conditioned on the input, the denoising blocks reconstruct the input. This does not yield a standard representation like other models that can be used for deterministic tasks. The authors have demonstrated classifcation performance on CIFAR but I think this is not sufficient. I would really like to see whether the approach works for ImageNet-1k as well as downstream dense prediction tasks such as semantic segmentation. Due to the time limit of the rebuttal, partial results are ok, for example, training only 100 epochs instead of 300 (standard ImageNet-1k training setting).
>
> First, we would like to clarify that in classification tasks **DiffusionBlocks do not reconstruct the input image $x$**. All blocks, including the last block, are optimized to predict the target class label $y$. Therefore, our transformation of the original network into the DiffusionBlocks does not change **the original problem of predicting class labels**. We only enable the block-wise training while maintaining comparable prediction performance.
>
> To reiterate, unlike the “diffusion models” in the usual CV applications, the subject being noised and denoised is the label embedding $y$, not the image $x$.
> As illustrated in the top-left panel of Figure 5, the label embedding $y$ is noised as $z=y + \sigma \epsilon$, and each block is trained to denoise $z$, conditioned on the CLS token and patches of the image $x$.
> At the inference time, we initialize $z$, an initial “noisy class label prediction” with Gaussian noise, and denoise it progressively through the blocks (see Figure 6, top left).
> This way, the denoised output of **each block** rightfully represents a class label prediction, and the total loss is defined as the weighted ensemble of the classification error from each block (Eq. (6)).
> For the evaluation, we use the “most clean” class label prediction $\hat{y}$ from the last block.
>
> As for the scale and diversity of the experiments, we would like to recall that although our main image classification experiment uses CIFAR-100, the language modeling experiments in Tables 3, 4, and 5 are also effectively large-scale classification tasks, where each step requires predicting one token out of a vocabulary of size 32,000.
>
> In hope to improve your confidence in our work, we also conducted additional experiments.
> Although we could not reserve the computational resource for ImageNet-1k within the limited period, we conducted a new experiment on the intermediate-scale, Tiny ImageNet dataset (64×64 resolution, 100k training images, 200 classes).
> We trained a 12-layer ViT from scratch (patch size 4, hidden size 768, 12 attention heads) for 100 epochs and obtained the following results:
>
> | Method | Accuracy (%, $\uparrow$) |
> | :--- | ---: |
> | ViT | 35.32 |
> | + DiffusionBlocks (B=2) | 36.16 |
>
> As we observed on CIFAR-100, LM1B, and OpenWebText, our modified network achieves competitive or slightly better performance than the standard end-to-end training.
> We have added this experiment to Appendix F.2. For reproducibility, we have also updated the Supplementary Material.
>
>
> We would like to emphasize that DiffusionBlocks applies broadly to diverse tasks (image classification, image generation, text generation) and model families (ViT, DiT, autoregressive transformers, masked diffusion models, and recurrent-depth models), and the output representation differs on tasks
>
> We appreciate the reviewer for motivating us to include this additional classification study.
>
> ----
>
> [5] Wu et al., Tiny ImageNet Challenge, 2017.
>
> ----
>
> Due to the character limit, we will address the remaining points in our fifth response.

---

> > ### Author Response · Authors · 2025-11-20
> >
> > ### 6. Use of the pre-trained models
> > > If I want to finetune a pretrained model on a dataset, can the weights and knowledge learned by the pretrained model ViT be used with this approach?
> >
> > This is indeed one of the most important future directions discussed in Section 6. The main challenge is that our diffusion-based objective in Eq. (6) is fundamentally different from the objective used for standard training. When a network is pretrained with an objective of different type, it remains uncertain if the pretraining benefits the new objective.
> >
> > The network architecture remains unchanged, and the conversion simply augments each block with noise conditioning and replaces the standard loss with the diffusion-based denoising objective (Eq. (6)). However, this introduces a significant optimization gap. Pre-trained models are optimized to minimize a task-specific loss computed from the final-layer output, whereas DiffusionBlocks trains each block independently to predict the target from noisy inputs. As a result, the learned representations in a pre-trained model may not transfer directly to the denoising objective, making straightforward fine-tuning more challenging than conventional end-to-end adaptation.
> >
> > Due to this mismatch in training objectives, we expect fine-tuning pre-trained models under the DiffusionBlocks framework to require additional strategies in order to fully leverage the pre-trained knowledge while enabling block-wise training. Developing methods to bridge this optimization gap, for example through hybrid objectives or progressive adaptation, is an important avenue for future research, and we view it as a promising direction for extending the practicality of DiffusionBlocks.
> >
> > ---
> >
> > We hope that these clarifications and additional analyses address your concerns and illustrate how DiffusionBlocks offers a principled and broadly applicable training framework.
> > We are grateful for your thoughtful feedback, which has helped us strengthen both the clarity and the overall quality of the work.

---

### Official Review · Reviewer_wyrV · 2025-10-30

**Soundness:** 2
**Presentation:** 3
**Contribution:** 3
**Rating:** 8
**Confidence:** 3

**Summary:**

The paper proposes a block-wise training strategy that splits standard transformer-based neural networks into groups of layers (blocks) which can then have forward/backward computed separately. This approach saved memory equal to the number of blocks, e.g. Bx memory savings. The paper achieves this by reconceptualizing the role of a block as a subnetwork that needs to denoise the output within a certain range. So, for a network with 12 layers spilt into 3 blocks, each group of 4 layers is responsible for removing a designated amount of noise, and the denoised output of the final set of 4 layers constitutes the actual prediction. The paper shows this method can compare to end-to-end training for image classification, image generation, and text generation.

**Strengths:**

S1. The method seems very novel. Converting these disparate tasks (classification, image generation, text generation) into denoising tasks and separating the forward/backprop for the network into different groups of layers (blocks) seems to solve a problem that prior block-wise training methods could not.

S2. Under the specified setups, the block-wise training matches or beats the baseline.

**Weaknesses:**

W1. On practical terms, this method would seem to have no advantage over recompute/checkpointing. In fact, the memory savings from recompute are substantially higher.

W2. It is not clear how this affects training efficiency (wall time).

W3. Baselines are odd. CIFAR-100 is a little toy-ish, and the default ViT performs quite poorly on it. The DiT is also under-optimized, and I assume this applies to other things as well. So it's not actually clear that this method can match any end-to-end training in practical setups, where these models are fully optimized.

**Questions:**

Since this method does not seem to be immediately practically useful, I am mainly viewing the strength in terms of the novelty (which I think is quite high). However, my mind could be changed (either for better or worse) depending on the following:

1. Can this be used effectively in tandem with recompute/checkpointing? What is the resulting impact on training time?

2. How long does this method take to train for ViT, DiT, etc.? Wall time preferred.

---

> ### Author Response · Authors · 2025-11-20
>
> We sincerely appreciate your insightful review and recognition of our method's novelty. Your practical concerns about DiffusionBlocks versus recompute/checkpointing are particularly valuable and have prompted us to provide crucial clarifications that will strengthen our paper's contribution to the community.
> For clarity, all modifications made in response to your comments are highlighted in $\textcolor{magenta}{magenta}$ in the revised manuscript.
>
> ### 1. Complementary Relationship with recompute/checkpointing
> > W1. On practical terms, this method would seem to have no advantage over recompute/checkpointing. In fact, the memory savings from recompute are substantially higher.
> > Q1. Can this be used effectively in tandem with recompute/checkpointing? What is the resulting impact on training time?
>
> We appreciate this important question about practical advantages. DiffusionBlocks and activation checkpointing offer fundamentally different trade-offs and can be powerfully combined.
>
> The key distinction lies in what each method reduces. Activation checkpointing reduces only activation memory, leaving parameters, gradients, and optimizer states unchanged. In contrast, DiffusionBlocks reduces all memory components by a factor of $B$. This distinction becomes increasingly critical as modern models grow larger.
>
> To illustrate this difference, consider an $L$-layer network where each layer has parameter size $P$ and activation size $A$. With Adam optimizer (requiring $2P$ for momentum and variance), each layer needs $4P$ memory for parameters, gradients, and optimizer states. Standard training thus requires $(4P + A)L$ total memory. Activation checkpointing reduces this to $4PL + A$ by rematerializing activations only when needed (though this is an optimistic estimate that ignores the memory cost of the checkpoints). DiffusionBlocks, by training $B$ independent blocks, requires $(4P + A)(L/B)$. Since $L > B$, combining DiffusionBlocks and activation checkpointing uses the least memory among these four patterns.
>
> Regarding computational costs, it is empirically known that activation checkpointing increases the training time by a factor of approximately 4/3, and this holds true when combined with the proposed method. This is justified as follows. With a forward pass computation cost of $F$, a backward pass requires approximately $2F$ (computing Jacobians and weight gradients). Standard training uses $3F$ cost per iteration, while activation checkpointing increases this to $4F$ due to recomputation. DiffusionBlocks maintains this ratio when combined with checkpointing.
>
> Beyond memory reduction, DiffusionBlocks offers unique advantages regarding training time: each block can be trained in an embarrassingly parallel manner. This means each block can be trained in parallel with absolutely no communication overhead. This provides an additional advantage over activation checkpointing, especially when computational resources are abundant.
>
> We have added these clarifications and the corresponding analysis to the revised manuscript in Appendix G, where we explicitly compare DiffusionBlocks with activation checkpointing in terms of both memory and
> computational cost, and describe how the two methods can be effectively combined.
> We appreciate the reviewer for highlighting this important practical consideration.
>
> ----
> Due to the character limit, we will address the remaining points in our second response.

---

> > ### Author Response · Authors · 2025-11-20
> >
> > ### 2. Training Efficiency and Wall Time Analysis
> >
> > > W2. It is not clear how this affects training efficiency (wall time)
> >
> > > How long does this method take to train for ViT, DiT, etc.? Wall time preferred.
> >
> > Thank you for raising this crucial practical consideration.
> >
> > From a computational perspective, standard end-to-end training processes an $L$-layer network for $K$ iterations, resulting in $K \times L$ layer computations. DiffusionBlocks trains only one block ($L/B$ layers) per iteration, so training all $B$ blocks for $K$ iterations results in $K \times (L/B) \times B = K \times L$ total layer computations, which is identical to standard training. The key difference is that DiffusionBlocks requires storing activations for only L/B layers at a time, yielding B-fold memory reduction without increasing the overall amount of computation.
> >
> > To validate this analysis, we measured wall time for a 12-layer ViT over 100 training iterations:
> >
> > - Standard training: 0.0507 [sec/iter] (for all 12 layers)
> > - DiffusionBlocks (B=3):
> >   - Per-block training time: 0.0181 [sec/iter] (for 4 layers)
> >   - Estimated total per-iteration time for all blocks: $0.0181 \times 3 = 0.0543 $ [sec/iter]
> >
> > The aggregated DiffusionBlocks time is therefore consistent with standard training, with the small difference attributable to noise-level conditioning (Section 3.1 Step 3). These measurements confirm that DiffusionBlocks does not introduce additional large computational overhead in practice.
> >
> > The key advantage emerges in multi-GPU settings. Since blocks train independently with no inter-block dependencies, DiffusionBlocks enables embarrassingly parallel training. Each of the B blocks can be assigned to a different GPU, training simultaneously without any communication overhead. This contrasts sharply with standard model parallelism or pipeline parallelism, which require frequent synchronization between GPUs. With B blocks on B GPUs, we can approach B$\times$ speedup compared to single-GPU training.
> >
> > We have added a detailed analysis of training efficiency and wall time to the revised manuscript in Appendix H.
> > ### 3. Baseline Strength and Practical Applicability
> >
> > > W3. Baselines are odd. CIFAR-100 is a little toy-ish, and the default ViT performs quite poorly on it. The DiT is also under-optimized, and I assume this applies to other things as well. So it's not actually clear that this method can match any end-to-end training in practical setups, where these models are fully optimized.
> >
> > We appreciate your concern about baseline strength. Our primary experimental objective is not to pursue SOTA performance, but to provide a fair and controlled comparison between DiffusionBlocks and standard end-to-end backpropagation. To isolate the effect of our training methodology from other factors such as pre-training, strong augmentations, we intentionally adopt from-scratch training with minimal augmentation across all settings.
> >
> > The seemingly low baselines therefore reflect this controlled setup rather than an inherent weakness of the architectures. For example, our ViT accuracy of 45% on CIFAR-100 is consistent with community reports under similar from-scratch training conditions [1], while accuracies above 80% typically rely on large-scale pre-training [2][3] or heavy data augmentation [4].
> >
> > To examine whether our findings hold beyond minimally optimized settings, we also conducted experiments with an enhanced training pipeline following [5]. Under this stronger setup:
> >
> > | Method | Accuracy (%, $\uparrow$) |
> > | :--- | ---: |
> > | ViT [5] | 59.6 |
> > | ViT (our impl.) | 60.25 |
> > | + DiffusionBlocks (B=3) | 59.30 |
> >
> > These results show that the relative performance relationship between DiffusionBlocks and end-to-end training remains stable across different optimization levels. Whether the baseline achieves 45% or 60%, DiffusionBlocks continues to match the baseline while providing substantial memory savings.
> >
> > We have included these additional experiments in the revised manuscript in Appendix F.2.
> >
> > ------
> >
> > [1] [Implementation of Vision Transformer from scratch and performance compared to standard CNNs (ResNets) and pre-trained ViT on CIFAR10 and CIFAR100.](https://github.com/ra1ph2/Vision-Transformer)
> >
> > [2] Dosovitskiy et al., An Image is Worth 16x16 Words: Transformers for Image Recognition at Scale, 2021.
> >
> > [3] Ridnik et al., ImageNet-21K Pretraining for the Masses, 2021.
> >
> > [4] Touvron et al., Training data-efficient image transformers & distillation through attention, 2020.
> >
> > [5] [Simple and easy to understand PyTorch implementation of Vision Transformer (ViT) from scratch, with detailed steps. Tested on common datasets like MNIST, CIFAR10, and more.](https://github.com/s-chh/PyTorch-Scratch-Vision-Transformer-ViT)
> >
> > ------
> >
> > We hope these clarifications address your practical concerns while reinforcing the novelty you appreciated.

---

> > ### Comment · Reviewer_wyrV · 2025-11-21
> >
> > Very helpful explanation, thank you.

---

> ### Comment · Reviewer_wyrV · 2025-11-21
>
> Helpful clarifications. I will follow the resolution of the other discussions before considering whether to raise my score. My initial concerns seem mostly resolved, and my primary lingering concern is that this won't scale to larger models, bigger data, more realistic training pipelines, etc. As a proof of concept, it seems solid as-is.

---

### Official Review · Reviewer_WpqB · 2025-10-31

**Soundness:** 2
**Presentation:** 2
**Contribution:** 3
**Rating:** 4
**Confidence:** 4

**Summary:**

This paper proposes to divide the network into K smaller subnetworks and train them independently.
In terms of architecture the paper assumes transformer-like residual networks, which are composed of a sequence of blocks.
A diffusion objective is used and noise level intervals are assigned to to each subnetwork.
During training, each subnetwork is trained alone and independently to denoise data in its assigned noise range.
The blocks are adapted so that their residual nature can be reframed as a denoising update of the form $z_{i+1}=z_i + \mu_{i+1} f(z_i)$, and in addition their input and ouput shapes should match the input data shape.
During inference, following the noise schedule, subnetworks for denoising are selected on the criteria that the current noise levels falls into their assigned noise range.
The proposed method makes both training and inference faster.
In addition this makes training scalable to larger architectures which as whole couldn't fit in current GPU memory but for which subnetworks can fit in memory.

The noise level assignment is done such that each network receives on average the same number of training noisified samples.

In addition, the method also targets recurrent-depth models.

**Strengths:**

1. The method looks simple and easy to implement.
2. The method targets an important problem: scalability and training/inference speed of large models.
3. For the most part, the paper is well written and easy to read.

**Weaknesses:**

1. However the notation sometimes lacks clarity, for example it's unclear what are $(x,y)$. From section 2.1, it appears that $y$ is data (e.g. an image for example). So that would make $x$ a label supposedly, however in Fig.3 $z_0\leftarrow x$ which suggest data rather than label. In Figure 2, in the case of the classifier, it would like $y$ is actually a label. A simple fix would be to clearly specify what $x$ and $y$ are and keep the notation consistent thorough the entire paper.
2. The results seem to compare to weak baselines, for example CIFAR100 classification baseline has an accuracy of 45% while SotA easily exceeds 80% if memory serves me well. Similarly DiT FID on CIFAR10 is reported at 39.83 while a google search reveals people obtaining FIDs within 10-15 on the DiT GitHub repo.
3. Nit, provide important information in the paper when possible, for example I had to dig into appendix D2 to find the resolution used for ImageNet (aka 256).
4. CIFAR10 is a surprising choice for benchmarking the method given that it's well known that CIFAR10 is subject to overfitting and uses heavy regularization whether in classification or diffusion. Therefore, given your method spreads the capacity over each subnetwork, it is unclear how much can be really explained by your $B$ ablation in table 8. I believe this ablation should be done on either ImageNet64 or 256.
5. On ImageNet256, the FID numbers are pretty high (10-12), it appears you DiT-L/2 while the DiT paper has most comparison for XL/2 (which gets an FID around 10), digging into the DiT paper appendix, the reported FID without CFG for DiT-L/2 is 23.33. I am confused, did you run DiT-L/2 or DiT-XL/2 as your results don't seem to match the ones reported by the original paper.

**Questions:**

1. In Fig.2 I am confused by your drawing, for step 3, you extract a block in the middle and refer to its output which is fed as an input with noise. What output are we talking about here, the actual output of the subnetwork or the final expected output of the whole network? From your algorithm in Fig.3 (the orange box), it appears it's the expected output $y$.
2. In 5.1 (ViT for image classification), you mentioned that noise is added to class label embeddings:
2.1 Where do class label embeddings come from, or are they really one-hot representations?
2.2 What type of noise is added to these embeddings? From Figure 2 it appears you add Gaussian noise to one-hot labels, is that right?
3. More generally, by slicing a network into K subnetworks, it stands to reason that each subnetwork has less capacity (given that the total number of weights remains about the same). How do you explain that your reported results in tables (1,2,3,4,5) are better than the full models they compare too while at the same much faster to train and with less capacity per subnetwork?
4. Why not also reporting performance with CFG on ImageNet256? Does your method works with CFG?

---

> ### Author Response · Authors · 2025-11-20
>
> We deeply appreciate your thorough and insightful review, particularly your attention to implementation details and practical considerations. Your careful analysis has identified important areas where we can strengthen our presentation. For clarity, all modifications made in response to your comments are highlighted in $\textcolor{magenta}{magenta}$ in the revised manuscript.
>
> ### 1. Notation clarity for $(x, y)$
> > the notation sometimes lacks clarity, for example it's unclear what are $(x, y)$. A simple fix would be to clearly specify what $x$ and $y$ are and keep the notation consistent thorough the entire paper.
>
> Thank you for pointing out this ambiguity. Indeed, $x$ denotes the conditioning/input data (e.g. image) and $y$ (e.g. class label) represents the target data to be predicted.
>
> The ambiguity in Figure 3 ($z_0 \leftarrow x$ in Standard Network Training) arises because the symbol $z$ is used in two different contexts:
>
> - $z_\ell:$ hidden states at layer $\ell$
> - $z_\sigma:$ noisy data at noise level $\sigma$
>
> In DiffusionBlocks, $\ell$’s layer input $z_{\ell-1}$ corresponds to $z_{\sigma_{\ell -1}}$ as established in Section 2.2 and Step 3 of Section 3.1.
> To help visualize this correspondence, Figures 5 and 6 (Appendix B) provide concrete examples across different architectures, illustrating how standard networks are converted to DiffusionBlocks.
>
> In response to your helpful feedback, we have revised the manuscript as follows:
> 1. We now explicitly define $x$ and $y$ at the beginning of Section 3.1.
> 2. We include a comprehensive notation reference table in Appendix A.
> ### 2. Baseline Performance
> > The results seem to compare to weak baselines, for example CIFAR100 classification baseline has an accuracy of 45% while SotA easily exceeds 80% if memory serves me well. Similarly DiT FID on CIFAR10 is reported at 39.83 while a google search reveals people obtaining FIDs within 10-15 on the DiT GitHub repo.
>
> Thank you for raising an important point about baseline performance.
> The main reason that our results looks weaker compared to SOTA results is because we intentionally design the experiments to isolate the effectiveness of DiffusionBlocks without relying on pre-trained models and heavy data augmentation etc.
> Importantly, our goal is not to achieve SOTA performance, but to ensure a fair comparison between end-to-end training and DiffusionBlocks under identical conditions.
>
> In fact, on CIFAR-100 classification, the above 80% accuracy typically comes from pre-trained models [1][2]. For from-scratch training, community implementations report similar performance to ours. For example, a 12-layer ViT (patch size 4) achieves 40.7% [3], comparable to our 45.26%.
> For DiT on CIFAR-10, the original paper [4] does not report CIFAR-10 results. The reported FID of 10-15 mentioned likely comes from [unofficial implementations](https://github.com/facebookresearch/DiT/issues/84) with unspecified model sizes, and computed FID against *training* distribution rather than the test set.
>
> To directly address your concern regarding baseline strength, we additionally conducted CIFAR-100 experiments using a enhanced training setting following [5]:
>
> | Method | Accuracy (%, $\uparrow$) |
> | :--- | ---: |
> | ViT [5] | 59.6 |
> | ViT (our impl.) | 60.25 |
> | + DiffusionBlocks (B=3) | 59.30 |
>
> These results show that DiffusionBlocks maintains comparable performance even under stronger training configurations, while still requiring only one-third of the memory.
> This confirms that our original design offers a fair comparison and that the findings are consistent across different training settings.
>
> These new results have been added to the revised manuscript in Appendix F.1. We sincerely appreciate your feedback, which helped us refine our experiments and improve the clarity of the presentation.
>
> ------
>
> [1] Dosovitskiy et al., An Image is Worth 16x16 Words: Transformers for Image Recognition at Scale, 2021.
>
> [2] Ridnik et al., ImageNet-21K Pretraining for the Masses, 2021.
>
> [3] [Implementation of Vision Transformer from scratch and performance compared to standard CNNs (ResNets) and pre-trained ViT on CIFAR10 and CIFAR100.](https://github.com/ra1ph2/Vision-Transformer)
>
> [4] Peebles et al., Scalable Diffusion Models with Transformers, 2022.
>
> [5] [Simple and easy to understand PyTorch implementation of Vision Transformer (ViT) from scratch, with detailed steps. Tested on common datasets like MNIST, CIFAR10, and more.](https://github.com/s-chh/PyTorch-Scratch-Vision-Transformer-ViT)
>
> ------
>
> Due to the character limit, we will address the remaining points in our second response.

---

> > ### Author Response · Authors · 2025-11-20
> >
> > ### 3. Experimental details in the main text
> > > Nit, provide important information in the paper when possible, for example I had to dig into appendix D2 to find the resolution used for ImageNet (aka 256).
> >
> > We thank the reviewer for pointing this out. In response, we have moved the key experimental details from the Appendix to the main text, including:
> >
> > * the ImageNet resolution (256x256) in Section 5.2
> > * the model architectures (DiT-S/2 for CIFAR-10, DiT-L/2 for ImageNet) in Section 5.2.
> >
> > These changes improve readability and allow readers to understand our experimental setup without needing to refer to the Appendix.
> >
> > ### 4. Ablation study validity on CIFAR-10
> > > CIFAR10 is a surprising choice for benchmarking the method given that it's well known that CIFAR10 is subject to overfitting and uses heavy regularization whether in classification or diffusion. Therefore, given your method spreads the capacity over each subnetwork, it is unclear how much can be really explained by your  ablation in table 8. I believe this ablation should be done on either ImageNet64 or 256.
> >
> > Thank you for pointing out this important consideration.
> > We used CIFAR-10 for ablations primarily because it enables systematic exploration of multiple design choices with manageable computational cost, which is essential for evaluating a broad set of configurations under controlled settings.
> >
> > The purpose of these ablations is to compare the *relative* effects of different design choices under strictly identical training pipelines, rather than to optimize absolute performance. Because every ablation variant uses the same data augmentation, regularization, and optimization scheme, any observed differences arise from the design choice being tested and not from dataset-specific effects.
> >
> > To examine whether the observed trends generalize beyond CIFAR-10, we conducted the same block-count ablation on ImageNet-256 using the DiT-L/2 architecture.
> >
> > | Blocks (B) | FID on CIFAR-10 ($\downarrow$) | FID on ImageNet-256 ($\downarrow$) |
> > |:---|---:|---:|
> > | 1 (baseline) | 39.83 | 12.09 |
> > | 2 | 35.47 | 9.90 |
> > | 3 | 38.03 | 11.11 |
> > | 4 | 45.43 | 11.90 |
> > | 6 | 53.32 | 14.43 |
> >
> > The consistent pattern across both datasets provides strong supporting evidence that the observed behavior of DiffusionBlocks is not an artifact of CIFAR-10. In particular, B=2 or B=3 achieves the best performance on both datasets, indicating that the insights from our CIFAR-10 ablations are robust and generalize to larger-scale settings.
> >
> > These new results have been added to the revised manuscript in Appendix F.3.
> >
> > ### 5. ImageNet-256 FID scores and CFG
> > > On ImageNet256, the FID numbers are pretty high (10-12), it appears you DiT-L/2 while the DiT paper has most comparison for XL/2 (which gets an FID around 10), digging into the DiT paper appendix, the reported FID without CFG for DiT-L/2 is 23.33. I am confused, did you run DiT-L/2 or DiT-XL/2 as your results don't seem to match the ones reported by the original paper.
> >
> > > Why not also reporting performance with CFG on ImageNet256? Does your method works with CFG?
> >
> > Thank you for carefully examining our ImageNet results. Our experiments use DiT-L/2 with classifier-free guidance (CFG) with scale 2.0 under the EDM framework. The discrepancy with the 23.33 FID reported in Table 4 of the DiT paper [4] arises from two important differences.
> >
> > 1. **Use of CFG:** The 23.33 score in the original DiT paper is the FID **without** CFG. Applying CFG is known to substantially reduce FID, which accounts for part of the gap.
> > 2. **Diffusion Framework:** Our experiments use the EDM framework [6], not DDPM as used in DiT. EDM provides improved noise schedules and training preconditioning that consistently yield lower FIDs than DDPM across various architectures.
> >
> > These two factors naturally bring the DiT-L/2 performance from the reported 23.33 (DDPM without CFG) to the 10-12 range observed in our experiments (EDM with CFG).
> >
> > Regarding your second question, DiffusionBlocks fully supports CFG. Each block is trained with class conditioning and 10 percent label dropout, following the standard CFG training recipe. This ensures that the blocks learn class-conditional denoising and can be used directly within classifier-free guidance at inference.
> >
> > For non-image generation tasks such as image classification and text generation, CFG is not natively supported in the corresponding baselines since these models are not diffusion architectures. Therefore, we did not use CFG for DiffusionBlocks in those settings in order to maintain a fair comparison.
> >
> > We have clarified in the revised manuscript (Section 5.2) that the ImageNet-256 experiments use **DiT-L/2 with CFG under the EDM framework**, which explains the difference from the scores reported in the original DiT paper.
> >
> > ----
> >
> > [6] Karras et al., Elucidating the Design Space of Diffusion-Based Generative Models, 2022.
> >
> > ----
> >
> > Due to the character limit, we will address the remaining points in our third response.

---

> > > ### Author Response · Authors · 2025-11-20
> > >
> > > ### 6. Output in Figure 2, Step 3.
> > >
> > > > In Fig.2 I am confused by your drawing, for step 3, you extract a block in the middle and refer to its output which is fed as an input with noise. What output are we talking about here, the actual output of the subnetwork or the final expected output of the whole network? From your algorithm in Fig.3 (the orange box), it appears it's the expected output $y$.
> > >
> > > Thank you for raising this important clarification. You are correct in your interpretation. The “output” in Step 3 refers to the target output $y$, not the intermediate output produced by a subnetwork. In DiffusionBlocks, each block is trained to predict the target data $y$ (such as a class label) given the input $x$ (such as image) together with a noisy version of the target $z_\sigma$.
> > >
> > > To clarify the training process:
> > > 1. Select a block $b$ with its assigned noise range $[\sigma_b, \sigma_{b-1}]$
> > > 2. Sample noise level $\sigma$ from this range and prepare noisy target: $z_\sigma = y + \sigma \epsilon$, where $\epsilon \sim \mathcal{N}(0, I)$
> > > 3. The block $b$ receives $(x, z_\sigma)$ and learns to recover clean target $y$ under its assigned noise range.
> > >
> > > This matches our training algorithm in Figure 3, where each block independently predicts $y$ from a noisy version of the target (see Line 4 in DiffusionBlocks - Training).
> > > Figure 5 in Appendix B also illustrates this more clearly across architectures by showing how noise is added to the target outputs (labels, embeddings, or images) during training.
> > >
> > > To remove the ambiguity, we have revised the caption for Figure 2 to explicitly state that Step 3 predicts the target $y$, ensuring that the intended meaning is unambiguous.
> > > We appreciate you pointing out this source of confusion and helping us improve the clarity of our presentation.
> > >
> > > ### 7. Label embeddings in ViT+DiffusionBlocks
> > >
> > > > In 5.1 (ViT for image classification), you mentioned that noise is added to class label embeddings: 2.1 Where do class label embeddings come from, or are they really one-hot representations? 2.2 What type of noise is added to these embeddings? From Figure 2 it appears you add Gaussian noise to one-hot labels, is that right?
> > >
> > > Thank you for these detailed questions about our ViT implementation. We clarify both points below.
> > >
> > > - **2.1 Class label embeddings:** The class label embeddings are learnable continuous vectors, not one-hot representations. We maintain an embedding layer $\mathbf{E} \in \mathbb{R}^{C \times d}$ in the original architecture, where $C$ is the number of classes and $d$ is the embedding dimension. For class $c$, the label embedding is obtained as $\mathbf{y}_{\text{emb}} = \mathbf{E}[c]$ as the standard protocol.
> > > - **2.2 Type of noise:** We add Gaussian noise to this continuous embedding: $z_\sigma = y_{\text{emb}} + \sigma \epsilon, \quad \epsilon \sim \mathcal{N}(0, I)$. In other words, the noise is applied to the learned $d$-dimensional embedding, not to one-hot vectors.
> > >
> > > Figure 5 and 6 in Appendix B illustrate how DiffusionBlocks is adapted across architectures, including how label embeddings are incorporated into ViT.
> > >
> > > To prevent any confusion, we have revised the corresponding text in Appendix B to explicitly state that we use learnable label embeddings rather than one-hot vectors.
> > > We appreciate you highlighting this point, which improved the clarity of our description.
> > >
> > > ---
> > >
> > > Due to the character limit, we will address the remaining points in our fourth response.

---

> > > > ### Author Response · Authors · 2025-11-20
> > > >
> > > > ### 8. Superior performance with specialized blocks
> > > > > More generally, by slicing a network into K subnetworks, it stands to reason that each subnetwork has less capacity (given that the total number of weights remains about the same). How do you explain that your reported results in tables (1,2,3,4,5) are better than the full models they compare too while at the same much faster to train and with less capacity per subnetwork?
> > > >
> > > > This is indeed a fascinating phenomenon that warrants deeper investigation and opens exciting research directions.
> > > > While the exact mechanism by which this gain occurs is a future work (as described in Section 6), we hypothesize several factors that may be contributing to the improved performance:
> > > >
> > > > 1. **Difference in objective functions:** Unlike standard end-to-end training where only the final layer’s outputs are used to compute the loss, each local block in DiffusionBlocks is linked directly to the target through a denoising objective in Eq.(6). This difference in learning signals may guide optimization more effectively.
> > > >
> > > > 2. **Specialization benefit:** As for the generation task, the original diffusion model uses a common network for each denoising step (Upper right of Figure 6). DiffusionBlocks, however, assigns smaller components for the denoising of different noise levels, which may lead to more efficient parameter utilization than a monolithic network handling all noise levels. Moreover, the block assignment comes with our particular block partitioning strategy (Section 3.3), which naturally implements a form of curriculum learning [7].
> > > >
> > > >
> > > > As noted in our Conclusion, fully understanding why structured optimization via diffusion can sometimes outperform end-to-end training is an important direction for future work.
> > > >
> > > > As an actionable improvement, we have also made the discussion of future work more explicit by separating it clearly in Section 6 of the revised manuscript.
> > > >
> > > > ------
> > > >
> > > > [7] Bengio et al., Curriculum learning, 2009.
> > > >
> > > > ------
> > > >
> > > > It is our hope that these responses further clarify the way that DiffusionBlock achieves memory-efficient training without compromising the final performance.
> > > > We are committed to incorporating all suggested improvements in the camera-ready version.

---

> > ### Comment · Reviewer_WpqB · 2025-11-24
> > **First pass reviewer response**
> >
> > 1. Acknowledged
> > 2. I understand and thanks for clarifying. I strongly believe using heavy augmentations should **not** be excluded in the name of fairness. You can apply heavy augmentations for all methods under comparison. The risk of low-accuracy baselines is that the network architecture capacity is left underused, possibly giving an unfair advantage to your method. The table you added with enhanced training is a lot more convincing to me than the earlier results (which I would recommend scratching from the paper for the capacity argument I raised above).
> >     1. "... computed FID against training distribution rather than the test set", actually that's the accepted norm in generative modeling to measure FID on the training distribution. Are you computing FID on the test set for your results?
> > 3. Acknowledged
> > 4. I understand the goal is to measure relative effects on CIFAR10, but given the overfitting nature of this dataset, it's hard to really trust that these effects would be the same on ImageNet64/256 for example. Since the table you share confirms that the effects are the same, I would advise to just report the one for ImageNet256 in the main paper and either discard the CIFAR10 one or move it to the appendix.
> > 5. Acknowledged
> > 6. Acknowledged - good.
> > 7. Thanks for clarifying the class embeddings come from a learnable weight matrix. I have a follow up question: since you add noise to these embeddings, what prevents the gradient descent from making the weights very large compared to the noise, effectively denoising them before they even enter the network?
> > 8. Acknowledged - very interesting.
> >
> > I'm willing to increase my score significantly if we can resolve the enhanced training / data augmentation argument for cifar100 (which you've shown you can do as the table you shared shows) and the capacity argument for cifar10 (which should be easy given you have ImageNet256 results now). To be 100% clear, by resolving I mean either convincing me otherwise or making changes to the paper.

---

> > > ### Author Response · Authors · 2025-11-27
> > > **Thank you for your response and further questions.**
> > >
> > > Thank you very much for your thoughtful follow-up feeback and questions.
> > > We fully agree with your points regarding 2. enhanced setting for CIFAR-100 and 4. ablation study for CIFAR-10. In response, we have made the following modifications to the revised manuscript (all highlighted in $\textcolor{magenta}{magenta}$):
> > >
> > > - **2. Enhanced training for CIFAR-100:** We have changed the main result of image classification task to the enhanced-training results in Table 1. We also applied the enhanced setting to Forward-Forward (FF) algorithm and reported its score (7.85) in Table 1. In fact, its accuracy decreases under the enhanced setting. Note that FF-trained model without enhanced learning was already underfitting to the dataset (Please see our original table). It is possible that, by applying even stronger regularization in the form of augmentation made the model underfit more. Furthermore, FF distinguishes positive and negative label assignments using only variations of the same image, so heavy data augmentations can substantially distort class-specific visual patterns, making this local discrimination task unstable.
> > >
> > > - **4. Ablation study on CIFAR-10:** Following your suggestion, we have updated Table 8. to the ImageNet-256 ablation, and the CIFAR-10 version has been moved to Appendix F.3.
> > >
> > > Below, we address your additional two questions.
> > >
> > > ------
> > >
> > > ### 2.1. FID evaluation: training vs test set
> > >
> > > > "... computed FID against training distribution rather than the test set", actually that's the accepted norm in generative modeling to measure FID on the training distribution. Are you computing FID on the test set for your results?
> > >
> > > Yes, our original experiments computed FID on the test sets (as described in Appendix E.2).
> > > Our intension was to follow the classic evaluation protocol of the machine learning, where the model is trained on the training data sampled from a target distribution and evaluated on unseen data from the same distribution.
> > >
> > > To ensure the full consistency with the accepted standard in image generation task, we additionally computed FID on the training sets following DiT [1] / ADM [2] evaluation protocol. The results are shown below.
> > >
> > > **CIFAR-10:**
> > > | Method | FID on test set | FID on training set |
> > > | :-- | --: | --: |
> > > | DiT | 39.83 | 32.84 |
> > > | **+ DiffusionBlocks** | **37.20** | **30.59** |
> > >
> > > **ImageNet:**
> > > | Method | FID on test set | FID on training set |
> > > | :-- | --: | --: |
> > > | DiT | 12.09 | 9.009 |
> > > | **+ DiffusionBlocks** | **10.63** | **9.004** |
> > >
> > > As shown, the relative trend between baseline and DiffusionBlocks is consistent across both evaluation protocols.
> > >
> > > We have added these training-set FID scores to the main Table 2 along with the test-set FID scores.
> > >
> > > ### 7.1. On the stability of embeddings under added noise
> > > > since you add noise to these embeddings, what prevents the gradient descent from making the weights very large compared to the noise, effectively denoising them before they even enter the network?
> > >
> > > Thank you for raising this insightful follow-up question.
> > > We apply L2 normalization to the embeddings following CDCD [3] to prevent the weights from growing very large relative to noise and also to avoid the collapse of the embedding space [3].
> > > We have explicitly described this process in Appendix C.
> > >
> > > ------
> > >
> > > We genuinely appreciate your detailed and constructive feedback. We hope our response has addressed your concerns. Please do not hesitate to raise any further questions about our technical analysis.
> > >
> > > [1] Peebles et al., Scalable Diffusion Models with Transformers, 2022.
> > >
> > > [2] Dhariwal et al., Diffusion Models Beat GANs on Image Synthesis, 2021.
> > >
> > > [3] Dieleman et al., Continuous diffusion for categorical data, 2022.

---

### Author Response · Authors · 2025-12-03
**Rebuttal Summary for the Area Chair**

Dear Area Chair,

We provide a concise summary of our rebuttal progress to help the AC understand the status. **We addressed every weakness and question** through detailed explanations, additional experiments, and revisions to the manuscript.

## Overall Summary

- Reviewer WpqB (Rating 4) responded once and provided specific actionable requests, explicitly stating that they would *“increase my score significantly”* if these were resolved. We completed all requested experiments and manuscript revisions just before the rebuttal was interrupted.

- Reviewer wyrV (Rating 8) responded positively (*“Very helpful explanation”*) and indicated they might raise their score after observing the outcome of the other discussions.

- Reviewer ZJo4 (Rating 6) did not send a follow-up reply, but we thoroughly addressed all points raised.

## Reviewer Highlights

### Reviewer WpqB (Rating 4)
This reviewer raised two major concerns, both of which we resolved through additional experiments and revisions. Importantly, the reviewer explicitly stated that they would significantly raise their score once these two issues were addressed.

**1. Baseline performance in Table 1 (CIFAR-100 classification):**
- The reviewer pointed out low accuracy in Table 1.
- We clarified that this arose from intentionally minimal augmentation to ensure fair comparison. (details in [2. Baseline Performance](https://openreview.net/forum?id=pwVSmK71cS&noteId=xcluRa2Dgl)).
- To directly address the concern, we conducted new experiments under an enhanced training setting, showing that DiffusionBlocks maintains comparable performance.
- Following the reviewer’s suggestion, Table 1 now reports the enhanced-setting results.

**2. Validity of the CIFAR-10 ablations:**
- The reviewer questioned whether CIFAR-10 reliably supports Table 8.
- We clarified the purpose of the ablation study and then performed additional ImageNet-256 ablations, which reproduced the same trends observed in CIFAR-10 (details in [4. Ablation study validity on CIFAR-10](https://openreview.net/forum?id=pwVSmK71cS&noteId=vnCzJjOR2h)).
- Following the reviewer’s suggestion, Table 8 now uses ImageNet-256 results.

The reviewer explicitly stated willingness to raise their score once these two issues were resolved, and we revised the manuscript accordingly.

### Reviewer wyrV (Rating 8)
This reviewer valued the novelty but asked about practical implications.

**1. Relation to activation checkpointing:**
- We provided an analytical comparison showing that DiffusionBlocks reduces *all memory components by a factor of $B$*, unlike activation checkpointing which reduces only activations. We also clarified that DiffusionBlocks can be combined with activation checkpointing (details in [1. Complementary Relationship with recompute/checkpointing](https://openreview.net/forum?id=pwVSmK71cS&noteId=PWS5XhCqSc)).
- The reviewer responded: *“Very helpful explanation, thank you.”*

**2. Training efficiency and wall time:**
- We reported actual wall-time measurements, showing only ~7% overhead from noise conditioning, while enabling embarrassingly parallel training across blocks (details in [2. Training Efficiency and Wall Time Analysis](https://openreview.net/forum?id=pwVSmK71cS&noteId=O0xqag9eVM).
- The reviewer replied that our clarifications were helpful and indicated no remaining concerns.

### Reviewer ZJo4 (Rating 6)
This reviewer highlighted the novelty and asked detailed questions about implementation and applicability. Although no follow-up reply was sent, we addressed all concerns.
Two points were particularly emphasized:

**1. Correspondence between ViT components and denoising steps:**

We clarified that the choice of which ViT component corresponds to a diffusion step is flexible; our implementation treats each ViT block as one step, but finer granularity is possible. (details in [2. Correspondence between ViT layers and denoising steps](https://openreview.net/forum?id=pwVSmK71cS&noteId=NAZAnznUPw)).

**2. Use of pre-trained models:**

We explained that this is indeed an important direction for future work described in Section 6, but a nontrivial direction due to the mismatch between training objectives. (details in [6. Use of the pre-trained models](https://openreview.net/forum?id=pwVSmK71cS&noteId=K9vrUo1gKv)).

----

## Final Remarks
Although the reviewers primarily focused on image-based experiments, our work demonstrates that DiffusionBlocks applies broadly: from ViT-based classification and diffusion-based image generation to masked language modeling, autoregressive text generation, and recurrent-depth models. This breadth shows the generality of the framework.

We hope that our extensive rebuttal, the additional experiments performed, and the reviewers’ expressed willingness to raise their scores had the rebuttal continued uninterrupted will be taken into account in your evaluation. We sincerely appreciate your time and consideration.

---

### Meta-Review · Area_Chair_UoTA · 2025-12-12

**Summary:**

This paper proposes a block-wise training strategy for transformer-style residual networks in diffusion models. The network is split into $K$ smaller subnetworks (groups of blocks), each assigned a specific noise-level interval and trained independently to denoise data. Because each subnetwork’s forward and backward passes are computed separately, the method reduces memory usage roughly by a factor equal to the number of blocks.

**Reviewer Concerns:**

The reviewers pointed out the questionable validity of the CIFAR-10 ablations on $B$ and suggested that the method should also be tested at the ImageNet 256 scale, which the authors have now addressed with additional experiments. They also raised concerns about whether the proposed method is truly memory-efficient and how it compares in terms of wall-clock time, both of which have been clarified with further results.

These additional experiments and clarifications are important for strengthening the paper. I recommend acceptance, and I strongly encourage the authors to incorporate these new results into the revised version.

**Reviewer Scores:**

The questions and concerns raised in particular by Reviewer WpqB appear to have been largely addressed, so I expect that the reviewer will increase their score.

---

### Decision · Program_Chairs · 2026-01-26

Accept (Poster)